

**De-risking the energy transition by quantifying the uncertainties in fault stability**
David Healy[1] & Stephen P. Hicks[2]
[1]School of Geosciences, University of Aberdeen, Aberdeen AB24 3UE United Kingdom
[2]Department of Earth Science and Engineering, Imperial College, London SW7 2AZ United Kingdom
d.healy@abdn.ac.uk
**Abstract**
The operations needed to decarbonise our energy systems increasingly involve faulted rocks in the
subsurface. To manage the technical challenges presented by these rocks and the justifiable public concern
over induced seismicity, we need to assess the risks. Widely used measures for fault stability, including slip
and dilation tendency and fracture susceptibility, can be combined with Response Surface Methodology from
engineering and Monte Carlo simulations to produce statistically viable ensembles for the analysis of
probability. In this paper, we describe the implementation of this approach using custom-built open source
Python code (pfs – *p*robability of *f*ault *s*lip). The technique is then illustrated using two synthetic datasets and
two case studies drawn from active or potential sites for geothermal energy in the UK, and discussed in the
light of induced seismicity focal mechanisms. The analysis of probability highlights key gaps in our knowledge
of the stress field, fluid pressures and rock properties. Scope exists to develop, integrate and exploit citizen
science projects to generate more and better data, and simultaneously include the public in the necessary
discussions about hazard and risk.
**Introduction**
*Rationale & Objectives*
Faults in the crust slip in response to changes in stress or pore fluid pressure, and the source of these changes
can be either natural or anthropogenic. Estimating the likelihood of slip on a particular fault for a given
change in loading is critical for the industrial operations of the energy transition, especially geothermal
energy and carbon sequestration and storage (CCS). The target formations of these operations are nearly
always faulted and fractured to some degree, and experience from waste-water injection in the USA shows
how even small changes in pore fluid pressure can trigger frequent seismic slip on these faults, with
significant and widespread impact on society (e.g., Elsworth et al., 2016; Hincks et al., 2018; Hennings et al.,
30 2019).

Stephenson et al. (2019) have shown how quantitative analysis of the subsurface is one of the key
contributions that geoscientists can make to decarbonising energy production to meet national and
international targets (e.g., CCC, 2019; IPCC, 2018). This includes the systematic geomechanical
characterisation of rock formations, better understanding of fluid flow in fractured rocks, and the need for
pilot projects to explore the scaling of behaviours from the laboratory to the field. Perhaps the most
important aspect is to understand the public attitudes to subsurface decarbonisation technology
(Stephenson et al., 2019; Roberts et al., 2021). Several recent studies have addressed the uncertainties in
subsurface structural analysis of faulted rocks (Bond, 2015; Alcalde et al., 2017; Miocic et al., 2019). In this
paper, we extend this work to specifically include fault stability, and argue that in order to simultaneously
address public concerns and assess the viability of different schemes, we need a more rigorous approach to
risking subsurface decarbonisation activities, especially where these involve changes in load on faulted rocks.
Useful measures of fault stability include slip and dilation tendency ($T_s$ and $T_d$ respectively) and fracture
susceptibility ($S_f$, the change in fluid pressure to push effective stress to failure). These measures are defined
as functions of the *in situ* stress, the orientation of the fault plane and, in the case of $S_f$, rock properties. It is
widely recognised that the inputs for the prediction of stability are always uncertain, and to varying degrees:
e.g., the vertical stress component of the *in situ* stress tensor can often be quite well constrained (to within



5%) from density log data, whereas the maximum horizontal stress is generally much harder to quantify. To
improve and focus our predictions of fault stability in the subsurface, we need to accept and incorporate
these uncertainties into our calculations. In this paper, we describe and explore a statistical approach to fault
stability calculations, and then apply these methods to examples in geothermal energy, in both low- and
high-enthalpy settings.
The specific aims of this paper are to:
1. describe and explain the Response Surface Methodology, and show how it can be applied to the
probabilistic estimation of fault stability using a range of different measures;
2. explore how the main variables – in situ stress, fault orientation and rock properties – relate to the different
measures of fault stability ($T_s$, $T_d$ and $S_f$) using synthetic (i.e., artificial) data;
3. use case studies of active and proposed geothermal projects with publicly available data to illustrate the
method, and then highlight the relationships between our known but uncertain input data and the predicted
risk of fault slip.
*Importance & Previous work*
Small changes in stress or fluid pressure (e.g., a few MPa) from human activities can have significant
consequences for fault stability. For example, waste-water injection from hydraulic fracturing ("fracking")
operations has led to dramatic increases in seismicity in Oklahoma since 2009 (Hincks et al., 2018) and in
Texas since 2008 (Hennings et al., 2019; Hicks et al., 2021). The precise mechanical cause(s) of this seismicity
is the subject of some debate, and could be due to either 'direct' pore fluid pressure transfer to basement-
hosted faults leading to a reduction in effective stress, or 'indirect' poroelastic effects at a distance (Elsworth
et al., 2016; Goebel et al., 2019). The concept of critically stressed faults in the crust (Townend & Zoback,
2000), where relatively high permeability serves to maintain near-hydrostatic pore pressures, is consistent
with the idea that only minor perturbations in loading can have dramatic consequences, even in areas of
apparently low seismicity and, implicitly, low background tectonic loading.
In densely populated areas such as the UK, public support for, and confidence in, subsurface operations are
key. Hydraulic fracturing operations for shale gas in Lancashire (UK) were stopped after earthquakes were
triggered by fluid injection (Clarke et al., 2019). Triggered felt seismicity has already been reported at the
United Downs deep geothermal pilot in Cornwall (Holmgren & Werner, 2021). Note that, in both of these
cases, fracturing and/or fault slip are intrinsic to the success of the operation as they are needed to enhance
fluid flow, and therefore earthquakes are inevitable. In detail, microseismicity (i.e., $M<2$) is inevitable, but it
is important to understand whether felt (i.e. $M>2$) seismicity can be forecast ahead of time. Furthermore,
many sites for energy transition projects in the UK are located in (beneath) areas of extreme poverty and
social deprivation, both rural (e.g., Cornwall, South Wales) and urban (e.g., Greater Manchester, Glasgow),
and therefore the risks from these projects fall disproportionately on the less well off (Nolan, 2016;
McLennan et al., 2019). To begin to address these complex issues, we need to quantify which faults are more
or less likely to slip in response to induced changes in loading. One approach is to analyse data during
subsurface operations and attempt to manage the consequences (e.g., Verdon & Budge, 2018). An
alternative approach, and the one taken in this paper, is to look at the bigger picture before operations
commence and reduce risk from the outset.
Various measures have been proposed to quantify the propensity or tendency of a given fault to slip (or
open) in a known stress field. The following methods are based around an assumption of Mohr-Coulomb
(brittle-plastic) failure which has been shown to capture the key aspects of faulting in the upper crust. Slip
tendency ($T_s$) was introduced by Morris et al. (1996) and is the simplest measure of fault stability, defined as:
$$T_s = \tau/\sigma_n \tag{1}$$
where $\tau$ is the shear stress and $\sigma_n$ is the normal stress acting on the fault plane. These stress components in
turn depend on the principal stresses and the orientation of the fault plane (see Lisle & Srivastava, 2004 for
details). In the absence of cohesion, if the slip tendency on a fault equals or exceeds the coefficient of sliding
friction, then the fault can be deemed "unstable". This dimensionless index embodies the key mechanical



principle underlying Mohr-Coulomb shear failure: as the shear ("sliding") stress acting on a fault plane rises
in relation to the normal (or "clamping") stress, the fault approaches failure and will slip. Dilation tendency
($T_d$) has been defined to describe the propensity for a fault to open, or dilate, in a given stress regime:
$$T_d = (\sigma_1 - \sigma_n)/(\sigma_1 - \sigma_3) \tag{2}$$
where $\sigma_1$ and $\sigma_3$ are the principal stresses of the *in situ* stress tensor (Ferrill et al., 1999).
Most rocks in the upper crust are porous and permeable to some degree, and fault rocks are no exception,
so these rocks are generally fluid saturated. This implies that we should include pore fluid pressure and the
concept of effective stress in our assessment of fault stability. Fracture susceptibility ($S_f$) is the change in pore
fluid pressure needed to push a stressed fault to failure (Streit & Hillis, 2004) and is defined by:
$$S_f = \Delta P_f = (\sigma_n - P_f) - (\tau - C_0)/\mu \tag{3}$$
where $P_f$ is the pore fluid pressure at the fault, $C_0$ is the cohesive strength (or cohesion), and $\mu$ is the
coefficient of sliding friction (see Figure 1b).

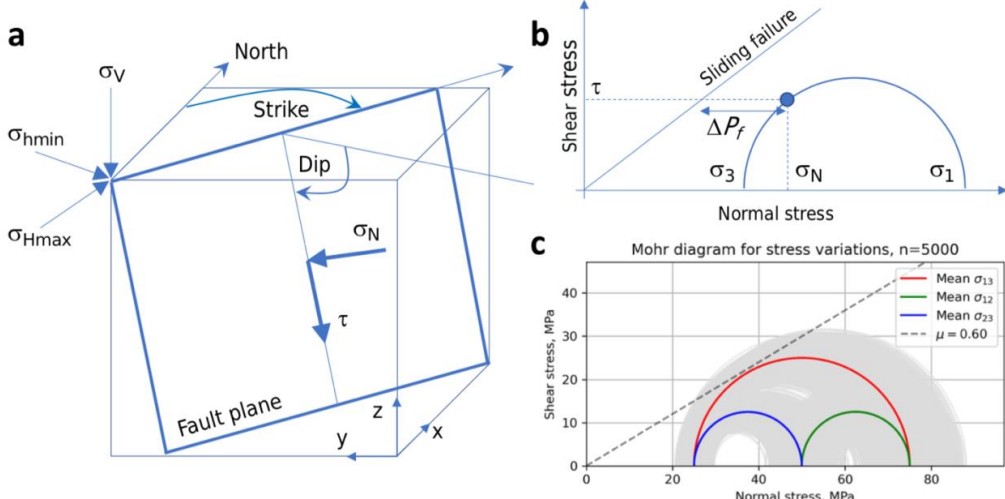


**Figure 1. a**. Schematic block diagram of a fault plane showing the terminology used in this paper. Also shown
are the cartesian and geographic reference frames and the Andersonian principal stresses. **b**. Mohr diagram
for a given state of stress (blue semi-circle) with normal ($\sigma_n$) and shear stresses ($\tau$) marked for a selected fault
plane orientation (blue dot). Failure envelope for frictional sliding (cohesion=0) also shown as straight blue
line. **c**. Mohr diagram depicting one of the key issues tackled in this paper: given uncertainty in the input
stress values (grey Mohr circles for the variation around the average principal stresses in red, blue and green),
what is the probability of failure? i.e., what percentage of all these stress states will intersect the failure
envelope?
Previous applications of these three measures of fault stability – $T_s$, $T_d$ and $S_f$ – cover the full spectrum of rock
types and stress fields, from basins to basement and from extensional, contractional and wrench tectonic
settings. Applications within the domain of the energy transition include examples from geothermal energy
(both shallow and deep) and CCS. The original definition of fracture susceptibility by Streit & Hillis (2004) was
concerned with safe injection limits for CO2 in potential reservoirs in Australia. Moeck et al. (2009) used slip
tendency to quantify the relative stability of different fault sets in different horizons in a geothermal reservoir
in the North German Basin, and Barcelona et al. (2019) used a similar method for Copahue geothermal
reservoir in Argentina. For CCS, Williams et al., (2016, 2018) have used slip tendency analyses of faults in
potential sandstone reservoirs on the UK continental shelf, including the North Sea and East Irish Sea basins.
The links between subsurface fluid flow, seismicity, and fault stability have recently been explored by Das &





Mallik (2020) for the Koyna earthquakes in India, and by Wang et al. (2020) for strike-slip faults in the Tarim
Basin of China.
Probabilistic approaches to fault stability have been adopted by various workers. In risking $CO_2$ storage for
an oil reservoir in the Williston basin, Ayash et al. (2009) used a features, events and processes (FEP)
approach to constrain the likelihood of occurrence of fault slip (based on slip tendency) and the severity of
the consequences, with their product defined as the risk. Rohmer & Bouc (2010) used RSM to assess cap rock
integrity for tensile or shear failure above deep aquifers in the Paris basin targeted for the storage of CO2.
Coupled RSM and Monte Carlo approaches to fault stability have been used by Chiaramonte et al. (2008) and
Walsh & Zoback (2016), following their initial application in the field of wellbore stability by Moos et al.
(2003). This Fault Slip Potential (FSP) method developed by Stanford (e.g., Chiaramonte et al., 2008 & Walsh
& Zoback, 2016) calculates the response surface for fracture susceptibility, with the in situ stress tensor
calculated by inversion of abundant seismicity data (focal mechanisms), and then uses a Monte Carlo
simulation to generate cumulative distribution functions (CDFs) of conditional probability of slip defined with
reference to an arbitrary pore pressure perturbation ($\Delta P_f$ = 2 MPa, in the case of Walsh & Zoback, 2016).
Note that FSP assumes cohesionless faults ($C_0$=0) and hydrostatic pore fluid pressure, and that *conditional*
probability in this sense refers to the fact that we do not know where any particular fault is with respect to
the seismic cycle.
*Conventions and layout for this paper*
In the sections below, we describe the underlying equations for measuring fault stability and then show how
we can use Response Surface Methodology (RSM) from engineering to explore the consequences of
uncertainties in the input variables. After assessing the quality of the solutions obtained from RSM, we then
apply a brute force Monte Carlo (MC) approach to generate cumulative distribution functions (CDFs) of the
different measures ($T_s$, $T_d$ and $S_f$). The case studies use published, publicly available data to constrain the
input variable distributions and then a combined RSM/MC approach is used to explore the uncertainty in
fault stability in different settings.
In this paper, compressive stress is reckoned positive, with $\sigma_1$ as the maximum compressive principal stress
and $\sigma_3$ as the minimum principal stress. Stress states and fault regimes are assumed to be Andersonian, with
one principal stress vertical, although the underlying model and code could be changed to incorporate non-
Andersonian stress states with the addition of extra variables for the stress tensor orientation (Walsh &
Zoback, 2016). The likelihood of slip on a fault is assessed in the framework of Mohr-Coulomb failure, with
or without cohesion (Jaeger et al., 2009). Fault orientations are quantified as strike and dip, following the
right-hand rule: with your right hand flat on the fault plane and fingers pointing down dip, the right thumb
points in the direction (azimuth) of strike. The relationship between the geographical and cartesian reference
frames follows a North-East-Down convention. Figure 1 depicts the key terms and elements used in the
analysis, and Table 1 contains a list of terms and symbols used with units where appropriate.

| Quantity | Symbol | Units |
|---|---|---|
| Maximum compressive stress | $\sigma_1$ | MPa |
| Intermediate compressive stress | $\sigma_2$ | MPa |
| Minimum compressive stress | $\sigma_3$ | MPa |
| Vertical stress | $\sigma_V$ | MPa |
| Maximum horizontal stress | $\sigma_{Hmax}$ | MPa |
| Minimum horizontal stress | $\sigma_{hmin}$ | MPa |
| Azimuth of max. horizontal stress | $sHaz$ | ° |
| Pore fluid pressure | $P_f$ | MPa |
| Coefficient of friction | $\mu$ | dimensionless |
| Cohesive strength (or cohesion) | $C_0$ | MPa |
| Slip tendency | $T_s$ | dimensionless |
| Dilation tendency | $T_d$ | dimensionless |
| Fracture susceptibility | $S_f$ | MPa |
| Fault strike | $\varphi$ | ° |



| Fault dip | $\delta$ | ° |
|---|---|---|
| Shear stress on a fault plane | $\tau$ | MPa |
| Normal stress on a fault plane | $\sigma_n$ | MPa |


**Table 1.** List of terms and symbols used in this paper, with units where appropriate.

**Statistical analysis of geomechanical fault stability**
*Introduction to Response Surface Methodology (RSM)*
RSM is widely used in engineering and industry along with a Design of Experiments approach, and often
employed to optimise a specific process of interest – e.g., to maximise the yield of a reaction given the input
variables of pressure, temperature, reactant mass etc. RSM is a large and growing field and is best considered
as a toolbox of different methods with a common mathematical basis. The governing equations for RSM were
derived by Box & Wilson (1951). The core idea is that a response *y* can be represented by a polynomial
function of a number (*q*) of input variables $x_1 - x_q$:
$$y = f\left(x_1, x_2, \dots, x_q\right) \qquad (4)$$
Each of the *q* input variables can be represented by either a discrete set of measurements made in the
laboratory (or field) or drawn from appropriate statistical distributions (normal/Gaussian, skewed normal,
Von Mises etc.). The simplest polynomial function that relates *y* and *x* is a linear one:
$$y_i = \beta_0 + \beta_1 x_{i1} + \beta_2 x_{i2} + \cdots + \beta_q x_{Nq} + \epsilon_i \qquad (5)$$
$$y_i = \beta_0 + \sum_{j=1}^{q} \beta_j x_{ij} + \epsilon_i \qquad (6)$$
where $\beta_q$ are the coefficients (to be determined), $y_i$ is the set of observations of the response (*i* = 1,2, ..., *N*),
and $x_{ij}$ are the input variables (*j* = 1,2, ..., *q*). $\epsilon$ is the experimental error, and the number of 'observations' *N*
> *q*, the number of input variables. This is therefore a multiple regression model linking the response *y* to
more than one (i.e., multiple) independent variables, *x*.
A more complex polynomial relationship is the quadratic form:
$$y = \beta_0 + \sum_{j=1}^{q} \beta_j x_j + \sum_{j=1}^{q} \beta_{jj} x_j^2 + \sum \sum_{i<j}^{q} \beta_{ij} x_i x_j + \epsilon \qquad (7)$$
This 2$^{nd}$ order multiple regression model contains all the terms of the linear (1$^{st}$ order) model, but also extra
terms for the squares and cross-products of the input variables (second and third terms on the RHS of
equation 7).
To define a response surface, either linear or quadratic, we need to calculate the values of the $\beta_q$ coefficients.
We can rewrite the key equations in matrix form:
$$\boldsymbol{y} = \boldsymbol{X\beta} + \boldsymbol{\epsilon} \qquad (8)$$
where $\boldsymbol{y}$ is an (*N* x 1) vector of observations (or calculations), $\boldsymbol{X}$ is an (*N* x *k*) matrix of input variable values (*k*
= *q* + 1), and $\boldsymbol{\beta}$ is a (*k* x 1) vector of regression coefficients. We solve this system of equations using the
standard linear algebra technique of least squares regression (Myers et al., 2016):
$$\widehat{\boldsymbol{\beta}} = (\boldsymbol{X'X})^{-1}\boldsymbol{X'y} \qquad (9)$$
The response surface (linear or quadratic) is then defined by
$$\widehat{\boldsymbol{y}} = \boldsymbol{X\widehat{\beta}} \qquad (10)$$
The values used in $\boldsymbol{X}$ are chosen to efficiently span the parameter space. A typical sampling design for $\boldsymbol{X}$ is
called the 3$^q$ model with 3 values of each variable, usually the minimum, mean (or mode) and maximum. For
slip tendency, *q* = 6 and this means we use 3$^q$ = 3$^6$ = 729 data points to calculate the response surface. In





practice, coded variables are used in **X** where the absolute values for the minimum, mean and maximum of
each variable are scaled to –1, 0 and +1 respectively, and then scaled back when the response surface is used
in the Monte Carlo simulation (Myers et al., 2016).
The response surface – i.e., the set of $\beta$ coefficients – is defined using a limited number of sample points,
depending on the chosen sample design ($3^q$ in the examples used in this paper; other variants exist – see
Myers et al., 2016 for details). To explore the possible variations of a response more fully, we use a Monte
Carlo (MC) approach of pre-defined size ($N_{MC}$ = 5,000 in the examples in this paper). The MC simulation uses
the response surface calculated from the design points to calculate the responses for $N_{MC}$ combinations of
input variables drawn from their distributions. This produces a statistically viable ensemble of response
values from which we can infer the probability of the response with respect to a chosen threshold.
With respect to fault stability, we can use RSM to produce a parameterised relationship – the response
surface in $q$ dimensions – between a stability measure of interest and the $q$ input variables. In the case of slip
tendency $T_s$, we can rewrite the components of equation 1 in terms of the measurable input quantities as
follows:
$$\tau = \sqrt{(\sigma_1 - \sigma_2)^2 l^2 m^2 + (\sigma_2 - \sigma_3)^2 m^2 n^2 + (\sigma_3 - \sigma_1)^2 l^2 n^2} \quad (11)$$

$$\sigma_n = \sigma_1 l^2 + \sigma_2 m^2 + \sigma_3 n^2 \quad (12)$$

where $l$, $m$ and $n$ are the direction cosines of the normal (pole) to the fault plane given by
$$l = \sin \delta \sin \phi \quad (13a)$$

$$m = -\sin \delta \cos \phi \quad (13b)$$

$$n = \cos \delta \quad (13c)$$

where $\phi$ is the fault strike and $\delta$ is the fault dip, in a North-East-Down reference frame (Allmendinger et al.,
220  2012).

All terms on the right-hand sides of equations 11-13 are uncertain to some degree, therefore estimating the
uncertainty of $T_s$, and as importantly, the *key controls on the uncertainty of $T_s$*, in terms of these input
variables, is non-trivial. This difficulty in estimating and visualising possible variations in our estimates of $T_s$
is exacerbated by the recognition that each of the input variables may be distributed differently: some
quantities (e.g., the principal stresses) may follow normal (Gaussian) statistics, whereas others (e.g., strike,
dip, sHmax azimuth) will follow Von Mises distributions. In the case of fracture susceptibility ($S_f$, equation 3),
it is even more complicated with the addition of three further input variables for friction, cohesion and pore
fluid pressure. Measurements or calculations of coefficients of friction and cohesive strength often display
asymmetric or skewed distributions (skewed high or low), and this adds further complexity to the task of
estimating and constraining fault stability from the data at hand.
*Worked Example 1: Slip tendency from synthetic input data*
The calculations presented in this paper were all performed with the custom pfs (**p**robability of **f**ault **s**lip)
package, written by the first author (DH) in Python 3, and freely available on GitHub (see Code Availability,
below).
The first example calculates a response surface for slip tendency ($T_s$) from $q$=6 input variables: the
magnitudes of the three principal stresses of the *in situ* stress tensor ($\sigma_1$, $\sigma_2$, $\sigma_3$) assumed Andersonian with
one principal stress vertical, the azimuth of the maximum horizontal stress (*sHaz*), and the strike and dip of
the fault plane. This response surface is then used in a Monte Carlo simulation ($N_{MC}$ = 5,000) to generate a
CDF of $T_s$ values for the fault. The specific Python code to run this example in the pfs package is wrapped in
a Jupyter notebook available on GitHub (WorkedExample1.ipynb).
The first task is to define the distributions of the input variables. In pfs, examples are shown for normal,
skewed normal and Von Mises (circular normal) distributions, but other statistical distributions are allowed.
Table 2 and Figure 2 describe the ranges and moments of these distributions for each input variable. For this





example, the normally distributed principal stresses are defined with a variation (standard deviation) of 5%
of their central (mean) value, and the Von Mises distributions of the azimuthal variables (sHaz, strike and dip)
all have $\kappa$ = 200 to model their dispersion about their mean. The fault of interest strikes 060° and dips 60° to
the south (right hand rule). The key questions to be addressed by this example are:

1.  given these uncertainties in the input stresses and orientation data, how does the estimation of $T_s$
    vary? What is the range and the mode?
2.  which variables exert the greatest (and least) control on the predicted variation in $T_s$?

We first build a response surface using a $3^q$ design, i.e., 3 data points for each variable – minimum, mean and
maximum – and for $T_s$, $q$ = 6. This means we calculate the response surface from $3^6$ = 729 data points. We
compare a calculated linear response surface with a quadratic response surface, using a normal probability
plot of residuals (Figure 3). These residuals are the differences between the values of $T_s$ derived from the
observations (taken from the input distributions shown in Table 2 (upper) and Figure 2), and the calculated
values of $T_s$ using the $\beta$ coefficients derived by least squares regression i.e., the response surface. The
adjusted $R^2$ value for the quadratic 2nd order model is significantly better than that for a linear 1st order model,
and we use quadratic models throughout the rest of this paper. More detailed inspection of the quality of fit
between the response surface and the observations is possible, including analysis of variance, main effects
plots and the use of t-statistics for each input variable to quantify their significance to the definition of the $\beta$
coefficients (Myers et al., 2016). In practice, visualising sections of the response surface for individual
variables is generally sufficient (see below; Moos et al., 2003; Walsh & Zoback, 2016).

| Variable | Mean | Standard deviation ($\kappa$ for Von Mises) | Units | Distribution | Comments |
|---|---|---|---|---|---|
| *Worked Example 1 – Synthetic $T_s$ – modelled depth=3 km* | | | | | |
| $\sigma_V$, vertical stress | 75.0 | 3.75 (5% of mean) | MPa | Normal | Lithostatic for depth of 3 km, assuming average rock density of 2500 kg m$^{-3}$ |
| $\sigma_H$, max. horizontal stress | 50.0 | 2.5 (5% of mean) | MPa | Normal | Andersonian normal faulting regime |
| $\sigma_h$, min. horizontal stress | 25.0 | 1.25 (5% of mean) | MPa | Normal | |
| Azimuth of $\sigma_{Hmax}$ | 060 | $\kappa$=200 | ° | Von Mises (circular Normal) | |
| Fault strike | 060 | $\kappa$=200 | ° | Von Mises (circular Normal) | |
| Fault dip | 60.0 | $\kappa$=200 | ° | Von Mises (circular Normal), truncated at 0 and 90 | |
| *Worked Example 2 – Synthetic $S_f$ – modelled depth=3 km* | | | | | |
| $\sigma_V$, vertical stress | 75.0 | 7.5 (10% of mean) | MPa | Normal | Lithostatic for depth of 3 km, assuming average rock density of 2500 kg m$^{-3}$ |
| $\sigma_H$, max. horizontal stress | 55.0 | 5.5 (10% of mean) | MPa | Normal | |
| $\sigma_h$, min. horizontal stress | 35.0 | 3.5 (10% of mean) | MPa | Normal | |





| $P_f$, pore fluid pressure | 30.0 | 3.0 (10% of mean) | MPa | Normal | Hydrostatic for depth of 3 km, assuming fluid density=1000 kg m⁻³ |
|---|---|---|---|---|---|
| Azimuth of $\sigma_{Hmax}$ | 060 | $\kappa$=200 | ° | Von Mises (circular Normal) | |
| Fault strike | 060 | $\kappa$=200 | ° | Von Mises (circular Normal) | |
| Fault dip | 60.0 | $\kappa$=200 | ° | Von Mises (circular Normal), truncated at 0 and 90 | |
| Friction, $\mu$ | 0.6 | 0.12 (20% of mean) | n/a | Skewed normal | $\alpha = -3$ i.e., skewed low |
| Cohesion, $C_0$ | 20.0 | 2.0 (10% of mean) | MPa | Skewed normal | $\alpha = +3$ i.e., skewed high |


**Table 2.** Table of input variable distributions for the synthetic models in Worked Examples 1 and 2.

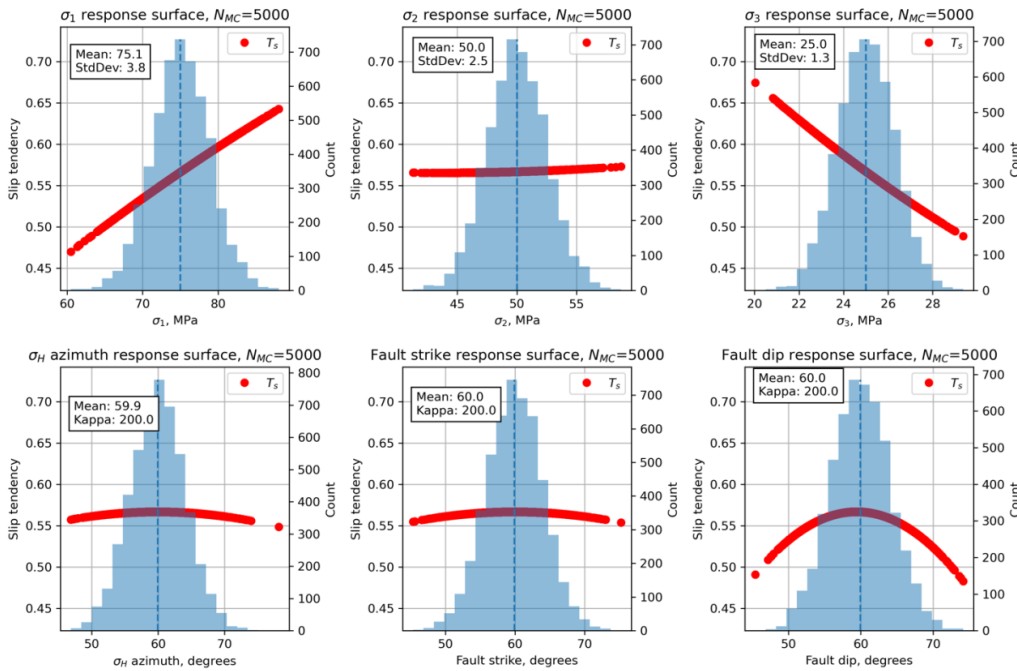


**Figure 2.** Histograms of input variables used to calculate slip tendency $T_s$ for the synthetic distributions shown
in Table 2.





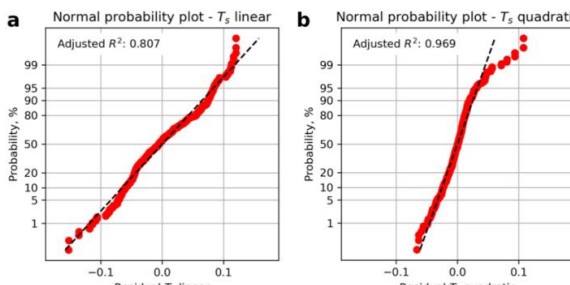


**Figure 3.** Residual plots for linear and quadratic response surfaces for slip tendency using synthetic data. The quadratic fit has a higher value of the adjusted $R^2$ parameter and is therefore deemed better in this case.

Having generated the quadratic response surface for $T_s$ for these input distributions, we can now use it to perform a Monte Carlo (MC) simulation with the aim of generating a statistically viable ensemble from which we can infer the probability of $T_s$ exceeding a critical value of sliding friction. The results from the MC analysis of $T_s$ are shown in Figure 4. The histogram of all values of $T_s$ shows a symmetrical and rather narrow distribution with a modal value of about 0.56 (Figure 4a). The CDF of all values of $T_s$ also shows this narrow and symmetrical distribution (Figure 4b).

A response surface of more than two variables is not easy to visualise. One approach is to take sections through the surface at specific values of all but one variable and graph that. The red lines shown in Figure 2 depict the response surface for that variable with all other variables held at their mean values. Thus the red line in Figure 2a shows the variation in $T_s$ as $\sigma_1$ varies with all other variables ($\sigma_2$, $\sigma_3$, $sHaz$, $\varphi$ and $\delta$) held at their mean values. There is a clear positive correlation of increasing $T_s$ with increasing $\sigma_1$, as expected from the definition of $T_s$ and its underlying dependence on differential stress (=$\sigma_1 - \sigma_3$); the clear negative correlation of $T_s$ with $\sigma_3$ shown in Figure 2c confirms this. Many of the response surface sections shown in Figure 2 are quasi-linear, but some are not: in particular, the dependencies of $T_s$ on $sHaz$, strike and dip are all non-linear, and this further justifies the selection of a 2$^{nd}$ order quadratic response surface model.

A useful way to visualise the results from the response surface calculated by the MC simulation is the tornado plot shown in Figure 4c. Here the ranges of $T_s$ for each input variable (shown as red lines over the histograms in Figure 2) are plotted to show the relative sensitivity of $T_s$ to each variable. Variables are ranked from the largest range at the top to the lowest range at the bottom. Again, the core dependence of $T_s$ on differential stress (=$\sigma_1 - \sigma_3$) is apparent, with $\sigma_1$ and $\sigma_3$ ranked highest in the plot. Interestingly, fault dip is ranked the next highest in terms of sensitivity and this reflects the geometry of this particular example. The Andersonian stress regime is for normal faulting, with $\sigma_1$ vertical. $\sigma_2$ is oriented parallel to fault strike (sHaz = strike = 060), and the fault dips at 60. This fault is therefore ideally oriented for slip in this stress field. Small changes to dip will influence the ratio of $\tau$ to $\sigma_n$, and therefore $T_s$.

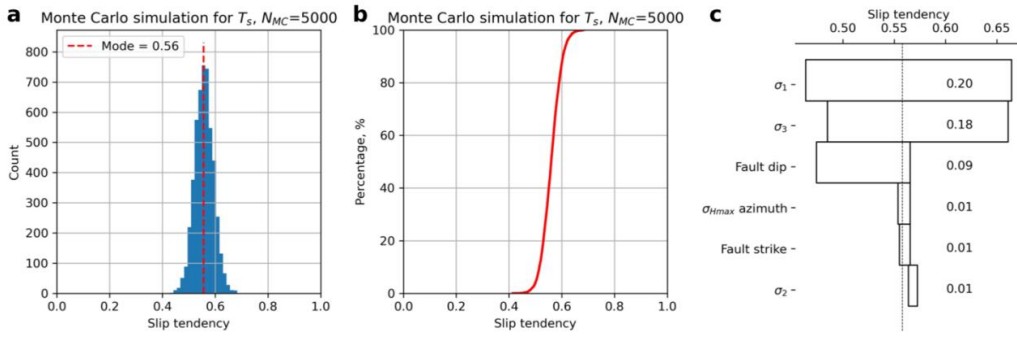




**Figure 4.** Output from Monte Carlo simulation ($N_{MC}$=5,000) of slip tendency calculated using a quadratic
response surface from synthetic input data. **a**. Histogram of calculated slip tendency values, in this case
showing a quasi-normal distribution with a mode of ~0.55. **b**. Cumulative distribution function (CDF) of
calculated slip tendency values, showing the range in values from ~0.4 to ~0.7. **c**. Tornado plot showing
relative sensitivity to the input variables. The vertical dashed line shows the modal (most frequent) value of
$T_s$ from the MC ensemble.

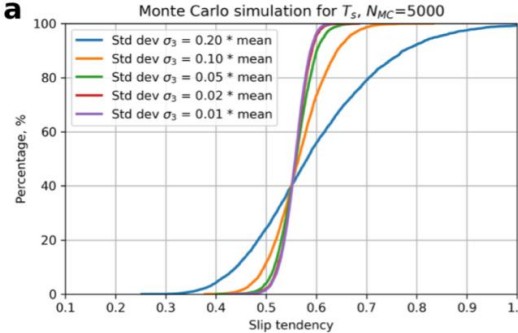

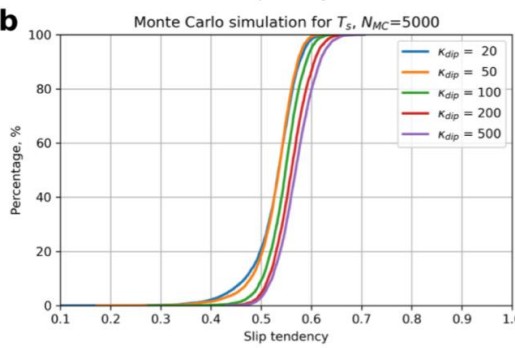


**Figure 5.** Output from Monte Carlo sensitivity tests for slip tendency, $T_s$. **a**. Effect of variation in standard
deviation of the least principal stress, $\sigma_3$. **b**. Effect of variation in dispersion ($\kappa$ parameter of the Von Mises
distribution) of fault dip.
We can use a Monte Carlo approach to explore these sensitivities in more detail. Given the shape of the
response surface sections shown in Figure 2 and the ranking of variables in Figure 4c, we can quantify how
more or less variation in the inputs will affect the predicted $T_s$. Figure 5 shows the results of this sensitivity
analysis for $\sigma_3$ and fault dip. The most significant effect on the CDF of $T_s$ is produced by increasing the
variation in $\sigma_3$ to 20% of the mean. This level of uncertainty for the minimum stress is not unreasonable in
real-world scenarios (see Case Studies below). Increased uncertainty in $\sigma_3$ at this level leads to a ~20% chance
of $T_s$ being in excess of 0.7 ($p$ = 0.8 for $T_s$ <= 0.7 from Figure 5a). Increased uncertainty in fault dip is achieved
by varying the dispersion parameter $\kappa$ of the Von Mises distribution (lower values of $\kappa$ = more dispersed).
Very disperse distributions of fault dip with $\kappa$ = 20 only change $T_s$ by < 0.1.
*Worked Example 2: synthetic Sf*
We can explore variations in predicted fracture susceptibility using the same principles as for slip tendency,
but adjusted by incorporating three new variables as required by equation 3 – pore fluid pressure, friction
coefficient and cohesion (code in GitHub: WorkedExample2.ipynb). The number of variables $q$ is now 9, and
therefore the design space used to compute the response surface is $3^q = 3^9 = 19,683$ data points. In practice
this means a slower run-time, but still only takes a few minutes on a modern processor.
For this example, we use the same stress tensor as for the $T_s$ example, with $\sigma_1$ as the maximum principal
stress and vertical, i.e., an Andersonian normal fault regime for a depth of approximately 3 km. We constrain





the *in situ* pore pressure with a symmetrical normal distribution with a mean value of 30 MPa, which is
approximately hydrostatic for a depth of 3 km, and with a variation of 10% of this mean. Friction is
constrained by a skewed normal distribution with a mode of 0.56 and $\alpha$ = −3, i.e., skewed towards lower
values. This shape of distribution for friction coefficients is consistent with previous studies (e.g., Moos et al.,
2003; Walsh & Zoback, 2016) but is open to question (see Discussion). Similarly for cohesion, we use a skewed
normal distribution with a mode of 21 MPa and $\alpha$ = +3, i.e., skewed towards higher values again consistent
with previous work. These input variable distributions are documented in Table 2 (lower) and shown in the
histograms of Figure 6.

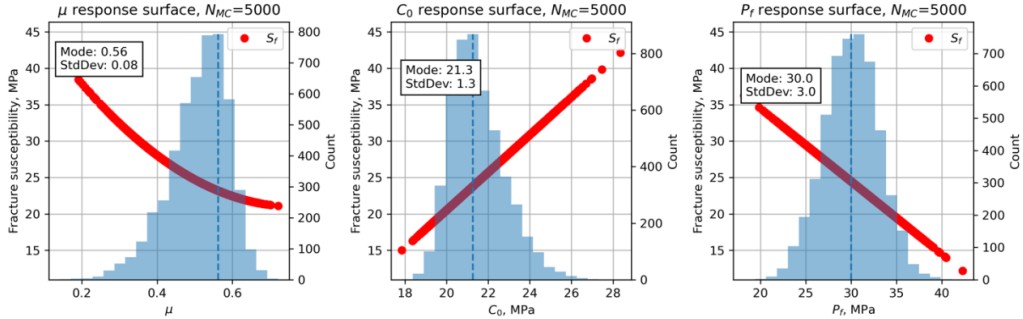


**Figure 6.** Histograms of the input variables, in addition to those shown in Figure 2, used to calculate fracture
susceptibility for the synthetic distributions shown in Table 2. Note the skewed (asymmetric) distributions
for $\mu$ and $C_0$.

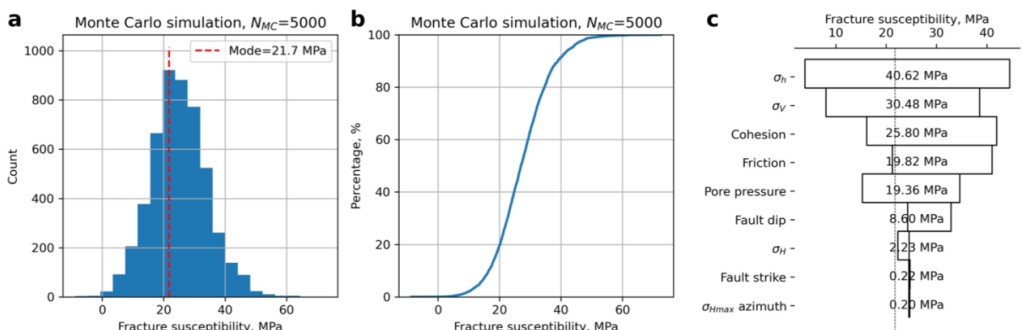


**Figure 7.** Output from Monte Carlo simulation ($N_{MC}$=5,000) of fracture susceptibility calculated using a
quadratic response surface from synthetic input data. **a.** Histogram of calculated fracture susceptibility,
showing a quasi-normal distribution with a mode of 21.7 MPa. **b.** Cumulative distribution function (CDF) of
calculated fracture susceptibility, showing the range in values from just less than 0 to about 60 MPa. **c.**
Tornado plot of relative sensitivities of the input variables used to calculate fracture susceptibility.

We calculate a quadratic response surface and use a Monte Carlo simulation ($N_{MC}$ = 5,000) to generate the
ensemble summarised in Figure 7. The mode of the distribution of $S_f$ is 21.7 MPa meaning that, on average,
an increase in pore fluid pressure of about 22 MPa above the average *in situ* value of 30 MPa is needed to
push the effective stress state to Mohr-Coulomb failure. The histogram in Figure 7a is approximately
symmetrical, perhaps with a slight skewness to higher values, and this is reflected in the CDF shown in Figure
7b. The distribution is overwhelmingly positive, meaning that this fault is almost unconditionally stable for
any change in pore fluid pressure, *at these conditions*. The response surface sections for $\mu$, $C_0$ and $P_f$ shown
in Figure 6 (red lines) all show a strong influence on the fracture susceptibility, and these are confirmed in
the tornado plot of Figure 7c. Pore fluid pressure exhibits a negative correlation with $S_f$ (Figure 6c) which is
consistent with the general principle of effective stress: i.e., if the original *in situ* pore pressure is already



high, it only takes a small perturbation (small $\Delta P_f = S_f$) to promote sliding failure. The response to changes in
$\mu$ and $C_0$ is more interesting (Figure 6a and b). For this magnitude of cohesion, the effect of cohesion on $S_f$ is
greater than that of $\mu$ ($C_0$ ranks higher than $\mu$ in the tornado plot, Figure 7c), and the dependence of $S_f$ on $\mu$
is negative. However, this relationship is not general as will be shown in the Case Study for the Porthtowan
Fault Zone (see below).

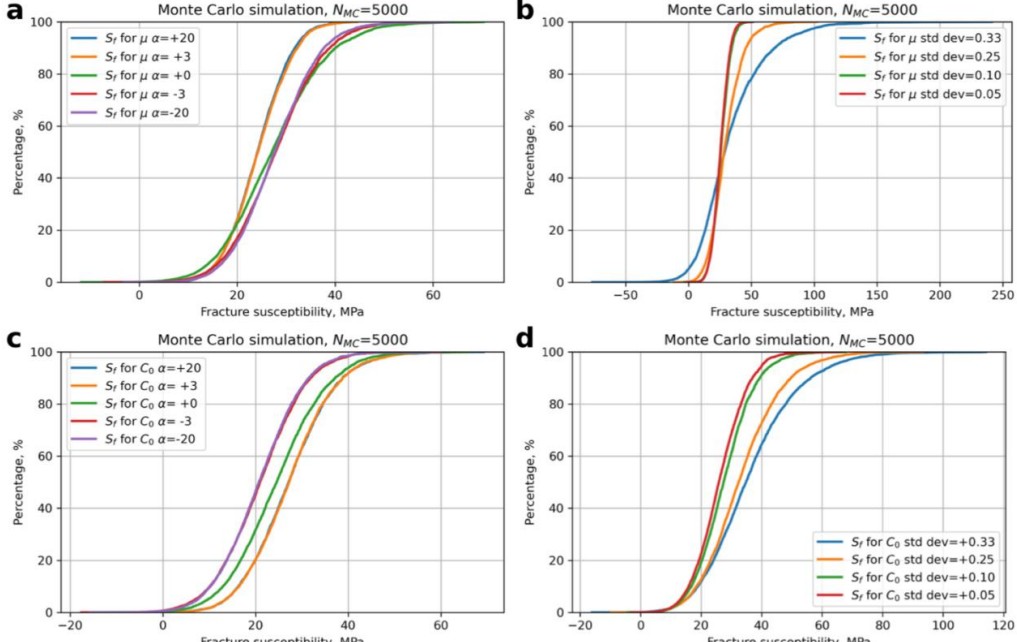


**Figure 8.** Sensitivity of fracture susceptibility to variations in $\mu$ and $C_0$. Note the changes in scale along the x-
axis between the plots.
The relative asymmetries of the skewed normal distributions for $\mu$ and $C_0$ have already been noted. Given
their significant effect on $S_f$ (high ranking in the tornado plot, Figure 7c), it is useful to explore how the
*skewness* of these distributions might influence Sf. Figure 8 shows the results of repeated Monte Carlo
sensitivity tests for $\mu$ (Figure 8a, b) and $C_0$ (Figure 8c, d). For friction, a positive skewness to higher values ($\alpha$
> 0) would tend to reduce $S_f$ – i.e., faults would be less stable. For cohesion, the opposite is true – a negative
skewness ($\alpha$ < 0) would make faults less stable to changes in $P_f$. These asymmetries are opposite to the ones
used in the main Worked Example 2 and used by other workers (see Discussion). Widening the distributions
for $\mu$ or $C_0$ by increasing their standard deviations (and retaining the original $\alpha$ values) tends to broaden the
distribution of predicted $S_f$ with asymmetry to higher (i.e., more stable) values.

**Case Studies**
The case studies have been chosen to illustrate how a combined RSM/MC approach can be used to estimate
the probability of slip on one or more faults, and to show that even with relatively good – i.e., complete –
input data, these predictions highlight that industrial operations remain significantly hazardous, with a
greater than 1 in 3 chance of slip on many faults across different settings. Selected specific aspects of the
modelling and the visualisation of results are emphasised in each case study. Figure 9 shows a map of the UK
with the case study areas marked, together with the locations of instrumentally-recorded earthquakes and
their focal mechanisms (Baptie, 2010). Also shown are data from the World Stress Map database of 2016
(Heidbach et al., 2018) indicating the orientation of the maximum horizontal stress. A basic observation from
this map is the level of complexity and heterogeneity in the present day seismotectonics of the UK, reflecting





the variation in the subsurface geology. However, there is a broad prevalence of NW-SE trending $\sigma_{Hmax}$
directions and strike-slip earthquake mechanisms.

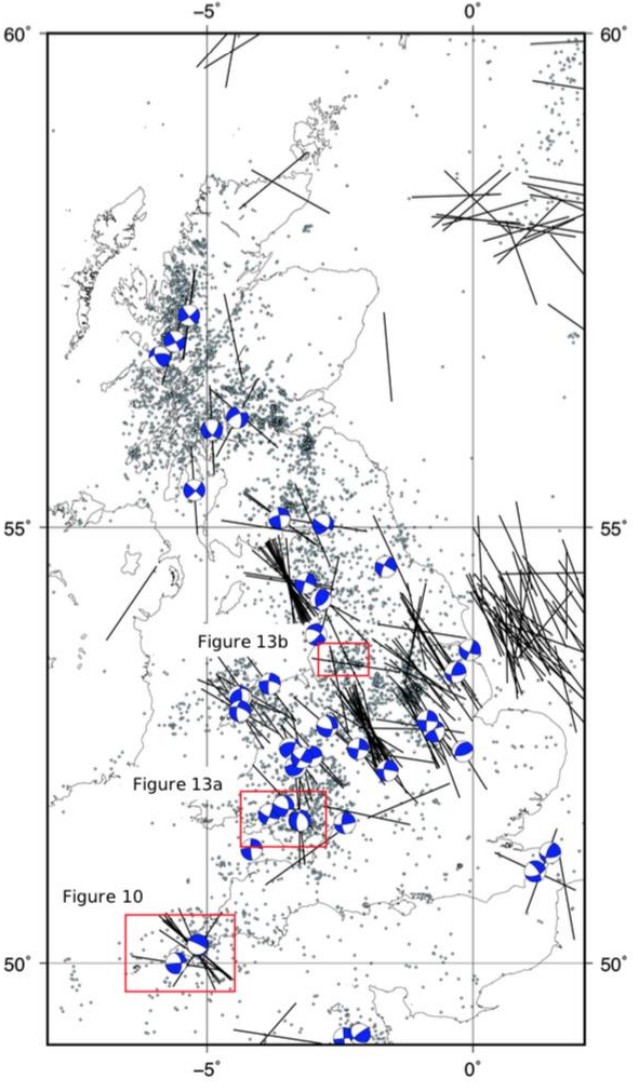

**Figure 9.** Map of most of the UK showing the locations of the selected case studies. Also shown: epicentres
of seismicity (light blue dots; BGS catalogue – Musson, 1996), focal mechanisms (blue and white; Baptie,
2010), and orientations of the maximum horizontal stress (black lines; World Stress Map data – Heidbach et
al., 2018).
*1.  Porthtowan Fault Zone in Cornwall, UK*
The Porthtowan Fault Zone (PFZ) cuts the Carnmenellis granite in Cornwall in southwest England (Figure 10).
This granite is a target for deep high-enthalpy geothermal energy due to its high radiogenic heat production
(Beamish & Busby, 2016). Following the Hot Dry Rock (HDR) project in the 1980s (Pine & Batchelor, 1984;
Batchelor & Pine, 1986), the United Downs pilot project has drilled two boreholes (UD-1, UD-2) to intersect
the fault zone at depths of about 5,275 m and 2,393 metres, respectively, making UD-1 the deepest onshore
borehole in the UK. The pilot project relies on shear-enhanced stimulation of pre-existing fractures (joints,



partially filled veins and faults) to drive fluid flow from the shallow injector (UD-2) to the deeper producer
(UD-1). Temperatures at the base of UD-1 have been predicted at about 200°C (Ledingham et al., 2019).
Shearing and downward flow of injected fluid was observed in boreholes as part of the earlier HDR project
and tracked with measured microseismicity (Pine & Batchelor, 1984; Green et al., 1988; Li et al., 2018).

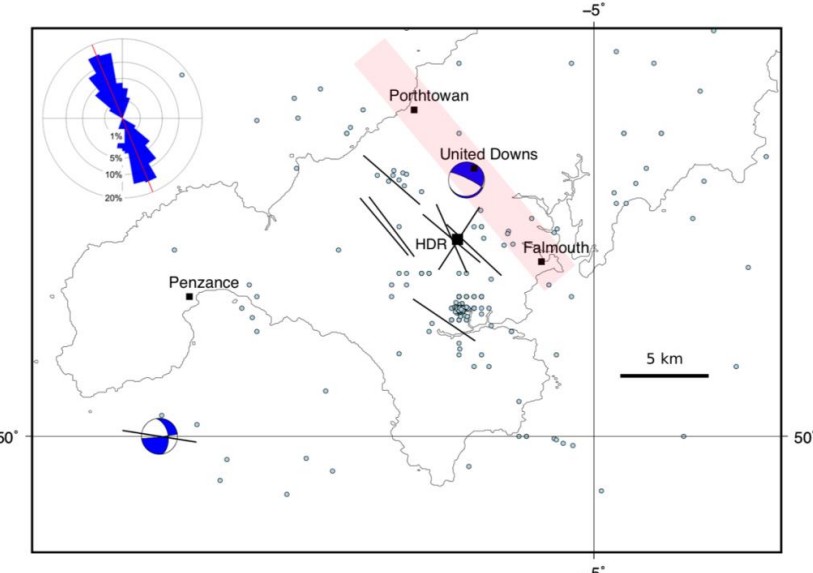


**Figure 10.** Map of South West England showing: selected population centres, the United Downs deep
geothermal pilot project and the former Hot Dry Rock project (black squares); epicentres of seismicity (light
blue dots; BGS catalogue – Musson, 1996); focal mechanisms (blue and white; Baptie, 2010); and orientations
of the maximum horizontal stress (black lines; World Stress Map data – Heidbach et al., 2018). Approximate
trend and extent of the Porthtowan Fault Zone shown in pale red. Inset shows an equal area rose diagram
with strikes of fault segments in the Porthtowan Fault Zone measured on BGS Falmouth sheet 352 (*N*=140;
circular mean strike=158°, circular standard deviation=27°).

Figure 10 shows a map of SW England overlain with seismicity data from the BGS (Musson, 1996). The PFZ is
poorly exposed inland, and runs NNW-SSE from Porthtowan on the north Cornish coast to Falmouth on the
south coast (see inset rose diagram for strikes of constituent faults taken from the BGS Falmouth sheet 352).
Overall, the fault zone is believed to dip steeply to the east at around 80°, but note that there is considerable
variation in strike and dip of individual fault and fracture planes within the fault zone (Fellgett & Haslam,
2021). The azimuth of the maximum horizontal stress is broadly NW-SE, with one exception trending NE-SW.


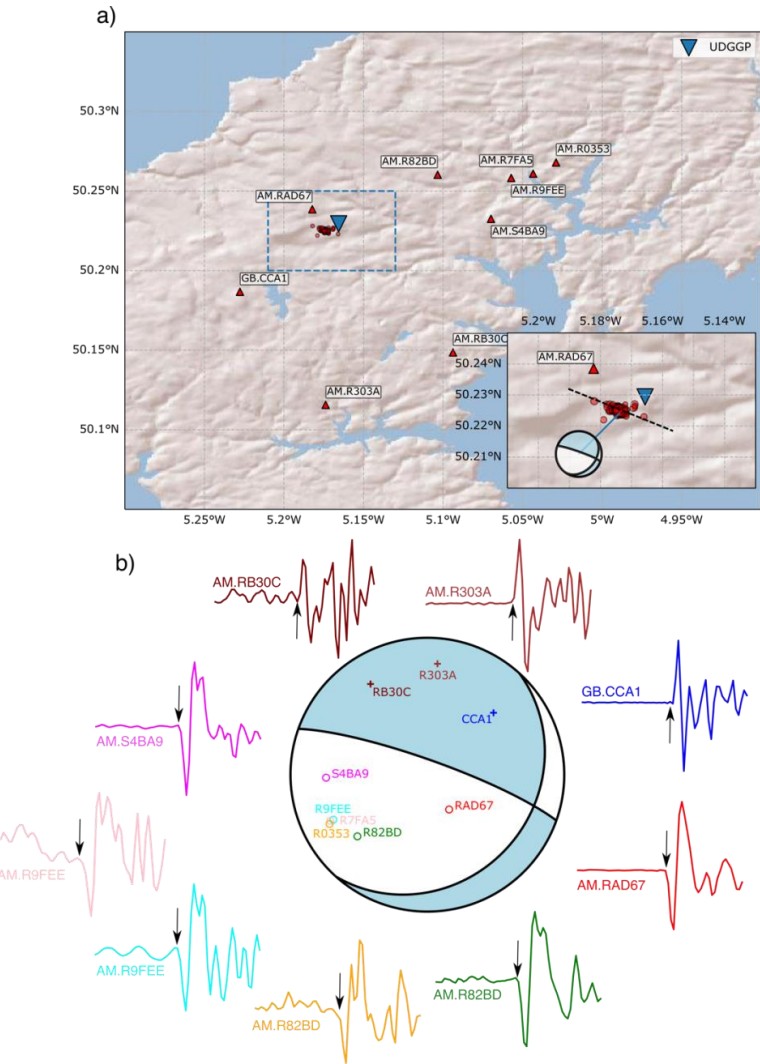


**Figure 11. a**. Red triangles show Raspberry Shake (network code: AM) and BGS (network code: GB) seismic stations in Cornwall, with station names labelled. Seismicity during geothermal operations is indicated by red circles. The inset shows a close-up of the area demarcated by the blue dashed line in the main map. The black dashed line in the inset shows the broad WNW-ESE alignment in seismicity. **b**. Computed focal mechanism for the 2020-09-30 11:44:01 $M_L$ 1.6 induced earthquake. First-motions are plotted on the focal sphere with "+" indicating positive polarity, and "o" for negative polarities. P-wave first-motions are plotted starting and ending 0.3 seconds before and after the picked arrival, respectively, and are coloured in the same way as the points on the focal sphere.

420



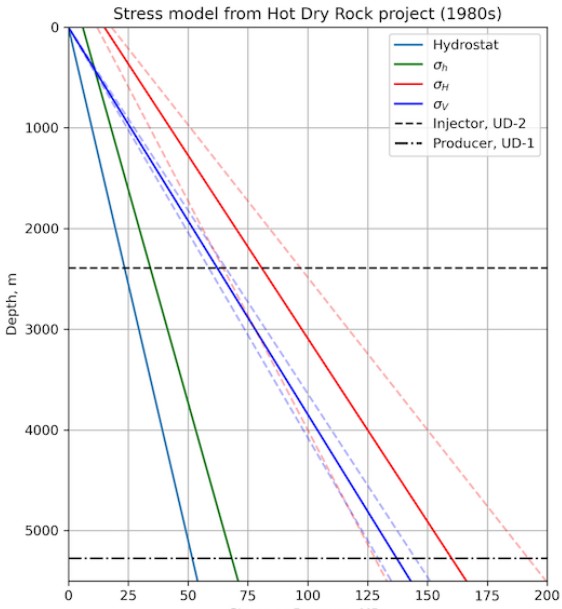

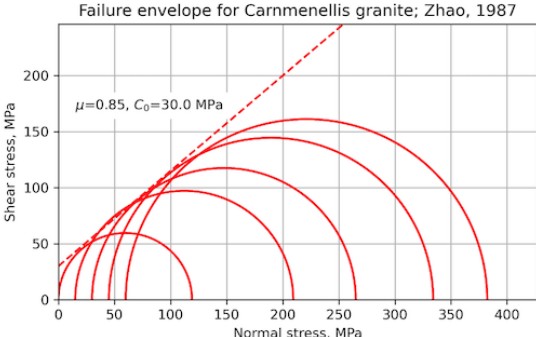

421

**Figure 12.** Constraints on input variables for the Porthtowan Fault Zone modelling. **a**. Stress-depth plot based
on data and equations from the Hot Dry Rock project in the Carnmenellis granite (Batchelor & Pine, 1986).
Also shown are the depths of the two wells in the pilot project at United Downs. **b**. Mohr diagram showing
data from laboratory mechanical tests of Zhao (1987) for brittle failure of Carnmenellis granite at 200°C.
Estimated Mohr-Coulomb failure envelope (dashed red line) is defined by $\mu$=0.85, $C_0$=30 MPa.

Detailed geomechanical analyses were performed in the Carnmenellis granite in the 1980s as part of the HDR
project, and these provide useful constraints on the variation of stress and fluid pressure with depth (Figure
12a; Batchelor & Pine, 1986). From these data, a strike-slip regime is most likely with $\sigma_1 = \sigma_{Hmax}$ and $\sigma_2 = \sigma_V$,
but note the uncertainties (based on quoted values in Batchelor & Pine, 1986): from around the depth of the
injector well at United Downs and deeper, a normal fault regime is also consistent with the data, i.e., $\sigma_1 = \sigma_V$
and $\sigma_2 = \sigma_{Hmax}$. Note that the earlier HDR project did not target a specific fault zone in the granite.

The thermo-mechanical properties of the Carnmenellis granite have been studied by Zhao (1987). Figure 12b
shows a Mohr diagram of data taken from Table 2.3 of Zhao (1987) for laboratory brittle failure tests
conducted at 200°C (the approximate temperature of the injector well at United Downs). From these data,
we have estimated a linear Mohr-Coulomb failure envelope defined by a friction coefficient of 0.85 and a





cohesive strength of 30 MPa. Cuttings from the boreholes at United Downs have been used to measure
friction coefficients of rocks within the PFZ, and values ranging between $\mu$=0.28-0.6 were recorded (Sanchez
et al., 2020).
We present model results for fracture susceptibility in the PFZ as the plan at United Downs (and elsewhere
in the future) is to inject fluid into the fault zone in order to generate shear-enhanced permeability on pre-
existing fractures. Table 3 lists the input variable distributions used in the "base case" model for hydrostatic
pore fluid pressure in the fault zone and mechanical properties taken from laboratory tests of intact
Carnmenellis granite (Figure 12b). The modelled depth is chosen as 4 km, in between the depths of the UD-
1 and UD-2 wells.

| Variable | Mean | Standard deviation ($\kappa$ for Von Mises) | Units | Distribution | Comments |
|---|---|---|---|---|---|
| $\sigma_V$, vertical stress | 105.0 | 5.25 (5% of mean) | MPa | Normal | Lithostatic for depth of 4 km, assuming average rock density of 2650 kg m$^{-3}$ Batchelor & Pine, 1986 |
| $\sigma_H$, max. horizontal stress | 125.0 | 25.0 (20% of mean) | MPa | Normal | Batchelor & Pine, 1986 |
| $\sigma_h$, min. horizontal stress | 53.0 | 5.3 (10% of mean) | MPa | Normal | Batchelor & Pine, 1986 |
| $P_f$, pore fluid pressure | 40.0 | 4.0 (10% of mean) | MPa | Normal | Hydrostatic for depth of 4 km, assuming average fluid density of 1000 kg m$^{-3}$ |
| Azimuth of $\sigma_{Hmax}$ | 140 | $\kappa$=200 | ° | Von Mises (circular Normal) | Batchelor & Pine, 1986 |
| Fault strike | 340 | $\kappa$=150 | ° | As mapped | Digitised from BGS map |
| Fault dip | 80.0 | $\kappa$=1000 | ° | Von Mises (circular Normal), truncated at 0 and 90 | |
| Friction, $\mu$ | 0.85 | 0.17 (20% of mean) | n/a | Skewed normal | $\alpha$ = −3 i.e., skewed low |
| Cohesion, $C_0$ | 30.0 | 6.0 (20% of mean) | MPa | Skewed normal | $\alpha$ = +3 i.e., skewed high |


**Table 3.** Distributions of input variables used in the base case model of fracture susceptibility in the
Porthtowan Fault Zone.



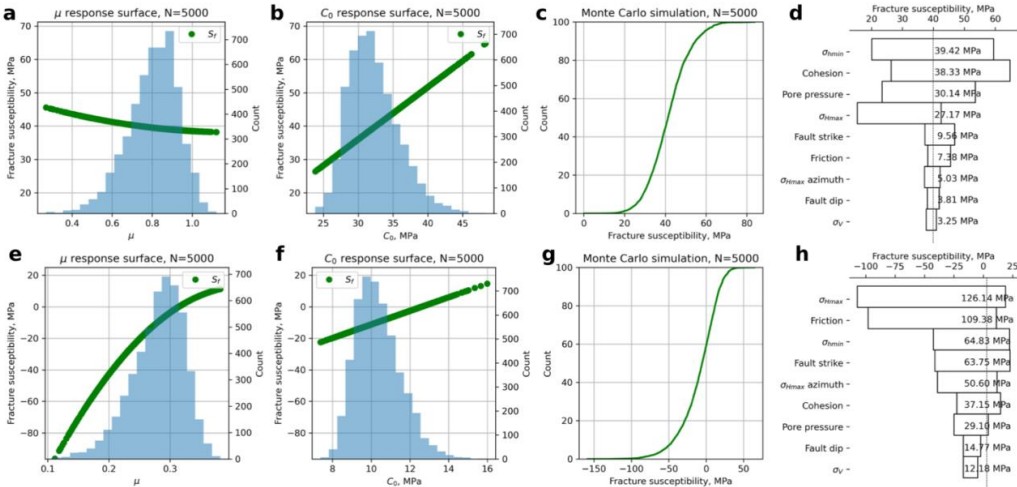


**Figure 13.** Outputs from the Monte Carlo simulation of fracture susceptibility in the Porthtowan Fault Zone.
**a-d.** The response surface for the base case, with friction and cohesion estimated from the laboratory failure
tests of Zhao (1987), predicts positive fracture susceptibility i.e., a stable fault zone. The tornado plot (**d**)
shows that for relatively high values of cohesion (mode of $C_0$=30 MPa in this case), the sensitivity to variations
in friction is slight. **e-h.** In contrast, the response surface for the 'weak fault' case, with reduced values of
friction and cohesion (mode of $\mu$=0.3, mode of $C_0$=10 MPa), predicts fault zone instability i.e., overwhelmingly
negative values of $S_f$. The effect of friction on these predictions is now very strong, as shown in the shape of
the response surface for $\mu$ (**e**) and in the ranking within the tornado plot (**h**).

The results from the Monte Carlo simulation of $S_f$ for the PFZ are shown in Figure 13. For the base case, with
hydrostatic pore fluid pressure and a 'strong fault' (mode of $\mu$=0.85, mode of $C_0$=30 MPa), the fault appears
unconditionally stable for the modelled *in situ* stress variations. The CDF shows almost exclusively positive
values of $S_f$ up to about 60 MPa. Note that, for the input stress variations listed in Table 3, 22% of the MC
simulations produced an Andersonian normal fault regime ($\sigma_1 = \sigma_V$), rather than a strike-slip ($\sigma_2 = \sigma_V$) regime.

232 microseismic events with hypocentre depths of 4-5 km were detected by the BGS during geothermal
testing operations in 2021-2022 (http://www.earthquakes.bgs.ac.uk/data/data_archive.html; last accessed
23 July 2021). The largest earthquake induced by geothermal operations during this period occurred on 2020-
09-30 11:44:01, and had a local magnitude of $M_L$ 1.6, and was felt by residents in the area. This event was
well-recorded on a network of single-component Raspberry Shake stations (e.g. Holmgren & Werner, 2021)
and a single station of the BGS permanent monitoring network (Figure 11a). These stations offer excellent
azimuthal coverage of the geothermal seismicity, with the closest station lying only 2 km away (AM.RAD67).
Since no focal mechanisms have yet been documented for these induced earthquakes, we used recorded P-
wave first motions to compute a focal mechanism of the $M_L$ 1.6 event using the method of Hardebeck &
Shearer (2002). Take-off angles were computed using a 1D seismic velocity model for the Cornwall area
(http://earthwise.bgs.ac.uk/index.php/OR/18/015_Table_4:_Depth/crustal_velocity_models_used_in_eart
hquake_locations; last accessed 23 July 2021). The best-fitting focal mechanism (Figure 11b) indicates either
normal faulting on a WNW-ESE steeply-dipping plane or strike-slip faulting on a shallow-dipping plane NE-
SW striking plane. Single event relocated epicentres reported by the BGS, which use arrivals from a local
dedicated microseismic monitoring array, show a NW-SE trend (Figure 11a), consistent with normal faulting
on a steeply east-dipping, WNW-ESE striking plane during this earthquake. Negative P-wave polarities were
recorded at AM.RAD67 for all $M > 0$ events, indicating that the same fault plane was reactivated during many
of the induced events. The inferred fault plane is sub-parallel to the interpreted strike of the Porthtowan
Fault Zone that is targeted by the geothermal testing. This observed normal faulting mechanism is consistent
with our MC simulations (more than 1 in 5 of the predicted stress states were for normal faulting).

483



The response surface (green lines on Figure 13a-b) and the tornado plot of relative sensitivities of the input variables (Figure 13d) shows a positive dependence of $S_f$ on the cohesion, and that variations in friction are relatively unimportant. If we reduce the strength of the modelled fault zone, by changing the input distributions of $\mu$ and $C_0$ to lower values – but with the same shape and skewness – the situation changes. The predicted fracture susceptibility is now much more strongly correlated with variations in friction, and less so with variations in cohesion. This can be explained by looking at the underlying formula for $S_f$ (equation 3), in particular the 2nd term on the RHS. If $C_0 > \tau$ then the numerator of this term can be negative, producing a net positive term. However, if $C_0 < \tau$ and $\mu$ is small then this term is larger and negative. The important point is that the probability distribution of $S_f$ (compare Figure 13c and 13g) is controlled by the *relative* magnitudes of $\mu$ and $C_0$. In a weak fault zone, with low $\mu$ and low $C_0$, the predictions are very sensitive to the value of friction. In a strong fault, the effect of $\mu$ is less important. Thus, we need to know more about the relationship between $\mu$ and $C_0$ in fault rocks (see Discussion).

2. *Coalfields in South Wales and Greater Manchester, UK*

Scope exists to extract low enthalpy geothermal heat from disused coalmines in the UK (Farr et al., 2016), using either open- or closed-loop technology. Possible sites include the South Wales and Greater Manchester coalfields, where folded and faulted Coal Measures of Westphalian (upper Carboniferous) age have been mined for centuries, up until the 1980s. Initial plans for shallow mine geothermal schemes include *passive* dewatering which may not change the loading on faults by much. However, *active* dewatering schemes can promote ingress of deeper ground water (Farr et al., 2021), and as this fluid flow must be driven by gradients in fluid pressure, this could in turn lead to the instability of faults at greater depth. The models below are for a depth of 2 km.

The locations and orientations of faults have been taken from published BGS maps. For the South Wales coalfield (Figure 14a), we used the BGS Hydrogeology map of S Wales to map the traces of faults in the Coal Measures (Westphalian), and BGS 1:50k solid geology sheets over the same area to collect data on fault dips. For the Greater Manchester coalfield (Figure 14b), we used the BGS 1:50k solid geology sheets for Wigan, Manchester and Glossop. Faults were traced onto scanned images of the maps in a graphics package (Affinity Designer on an Apple iPad using an Apple Pencil). These fault trace maps were saved in Scalable Vector Graphics (.SVG) format, after deleting the original scanned image layer of the geological map. The saved .SVG files were read into FracPaQ (Healy et al., 2017) to quantify their orientation distributions (inset rose plots in Figure 14a and b). The fault trace maps were then overlain on maps containing historical seismicity and available focal mechanisms (from the public BGS catalogue; Musson, 1996) and the orientations of $\sigma_{Hmax}$ taken from the World Stress Map project (Heidbach et al., 2018).

In the South Wales coalfield 3,408 fault segments were traced, and the dominant trend is clearly NNW/SSE, but with important (and long) fault zones running ENE-WSW, such as the Neath and Swansea Valley Disturbances (Figure 14a). From cross sections, we measured 142 fault dips to help constrain the distribution of friction coefficients in these rocks (Figure 15b-c; see below), corrected for vertical exaggeration on the section line where necessary. Focal mechanisms in this area (*n*=4) suggest that NNW/SSE and N/S faults are active in the current stress regime. Historical seismicity is widely, if unevenly, distributed with no obvious direct correlation to the surface mapped fault traces. For example, there are areas of intense surface faulting but no recorded historical seismicity, and vice versa – areas with abundant historical events but few mapped faults.

Around Greater Manchester 3,453 faults were traced, and the dominant trend is NW/SE, but E/W faults are also present (Figure 14b). From cross sections, we measured 89 faults to help constrain the distribution of friction coefficients in these rocks (Figure 15d-e; see below). Historical seismicity is again widely, if unevenly, distributed with few obvious direct correlations to the surface mapped fault traces. However, there was an earthquake swarm in 2002-2003 which comprised more than 100 events, with a maximum local magnitude of 3.9. Calculated focal depths were 1 – 3 km, although these have large uncertainties (Walker et al., 2003). The World Stress Map database has the orientation of $\sigma_{Hmax}$ trending WNW/ESE in this area (Figure 12b), based on the focal mechanisms for local events in the 2002-2003 swarm (this is distinct from the regional



trend of $\sigma_{Hmax}$ which is more NW/SW e.g., Williams et al., 2016). These observations suggest that faults
oriented more nearly E/W are more likely to slip in the current stress regime.
There are no published geomechanical analyses for the variation of stress with depth for either of these two
areas. To constrain the depth dependence of stress, we have used larger scale syntheses of stress for onshore
UK produced by the BGS (e.g., Kingdon et al., 2016; Fellgett et al., 2018). The stress-depth plot in Figure 15a
has been constructed using the data shown in Fellgett et al. (2018), and shows that, in general, a strike-slip
fault regime with $\sigma_1 = \sigma_{Hmax}$ is most likely. However, given the known uncertainties in these data, a normal
fault regime ($\sigma_1 = \sigma_V$) cannot be ruled out, especially at depth. Note that the stress-depth data shown in
Fellgett et al. (2018)  and used in Figure 15a are compiled from different areas, and remain untested for the
specific areas shown in this paper. The azimuth of $\sigma_{Hmax}$ is known to vary across the UK ranging from ~130 to
~170 (Baptie et al., 2010; Becker & Davenport, 2001).
Despite the economic and historical significance of the Coal Measures, there are no published datasets of
laboratory measured friction or cohesion for either intact rocks or their faulted equivalents (although data
may exist in proprietary company records). Data for specific units of interest does exist, e.g., for the
Oughtibridge Ganister, a seat earth in the Coal Measures (Rutter & Hadizadeh, 1991); and the Pennant
Sandstone, a rare marine sandstone unit (Cuss et al., 2003; Hackston & Rutter, 2016), but a systematic
analysis of the volumetrically dominant sandstone, siltstone and mudstone formations is notably absent.
Instead, we use the measured dips of faults in the Coal Measures as a proxy for the coefficient of sliding
friction, using the relationship
$$\mu = 1/\tan(\pi - 2\beta) \qquad \text{equation 14}$$
where $\beta$ is the angle between the fault plane and $\sigma_1$ at failure (Jaeger et al., 2009; Carvell et al., 2014). Such
a calculation assumes Mohr-Coulomb failure and that the current dip of the fault is reasonably close to the
dip at failure in the post-Westphalian deformation of the coalfields. For measured fault dips < 45°, we assume
that $\sigma_1$ was horizontal (Andersonian thrust/reverse fault regime) and for fault dips >= 45° we assume $\sigma_1$ was
vertical (Andersonian normal fault regime). In practice, some of these faults probably originated as strike-slip
faults (i.e., with a sub-vertical dip and $\sigma_2$ vertical), and some of their dips have almost certainly been modified
by compaction since their formation. However, this method of estimating the likely range of friction
coefficients from measured dips remains simple to apply and useful to first order, in the absence of better
data. From the dip data, the calculated friction coefficients vary between 0.0 and 6.0 for South Wales, and
between 0.35 and 2.0 for Greater Manchester (Figures 15c and e, respectively).


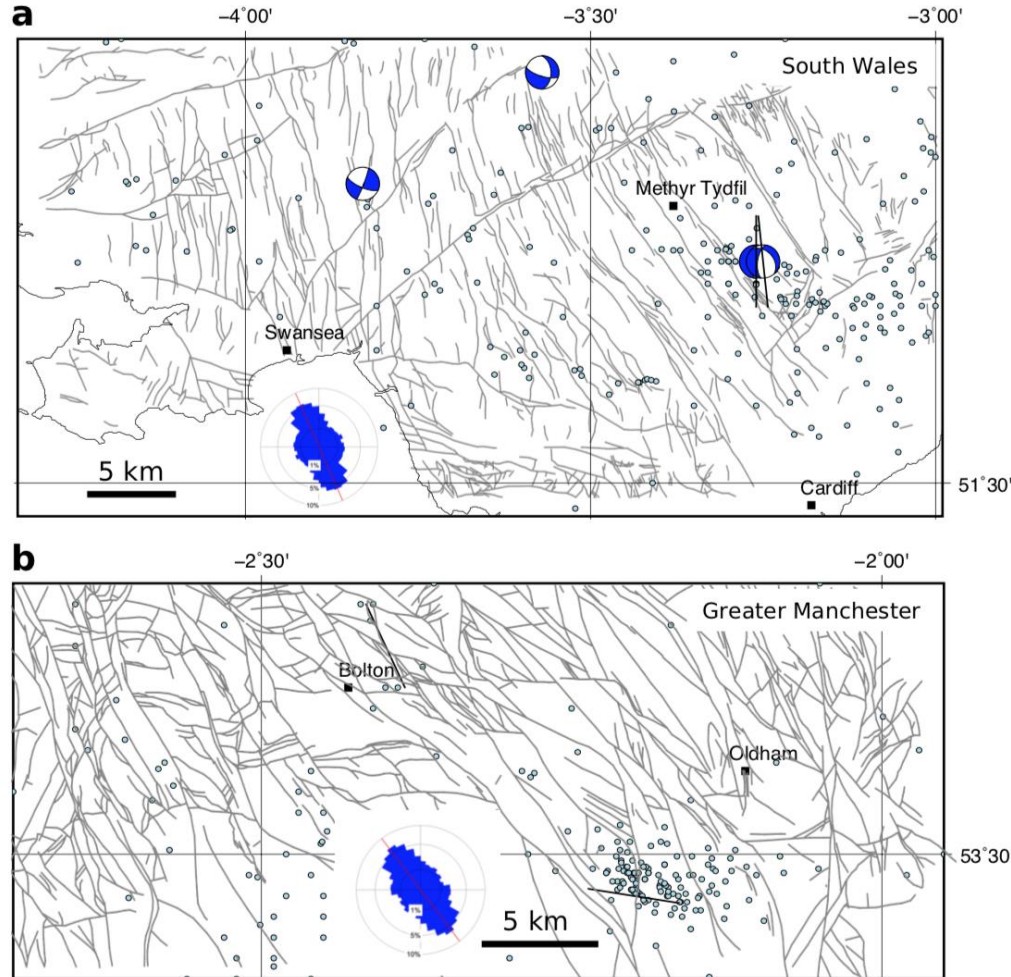


**Figure 14.** Maps of selected UK coalfields (suggested sites of shallow mine geothermal energy) showing: selected population centres (black squares); epicentres of seismicity (light blue dots; BGS catalogue – Musson, 1996); focal mechanisms (blue and white; Baptie, 2010); and orientations of the maximum horizontal stress (black lines; World Stress Map data – Heidbach et al., 2018). Inset equal area rose diagrams show orientations of mapped faults. **a**. South Wales area. Faults in the Coal Measures taken from the BGS Hydrogeological Map of South Wales (1:125k) (*n*=3,408), with a circular mean strike=156° and a circular standard deviation=65°. **b**. Greater Manchester area. Faults in the Coal Measures taken from the BGS 1:50k sheets Wigan, Manchester and Glossop (*n*=3,453), with a circular mean strike=143° and a circular standard deviation=64°.

Based on the values of sliding friction calculated from measured fault dips across both coalfields a threshold stability value of $\mu$=0.3 is taken as a reasonable lower bound for faulted rock. This is the value used to compare with predicted slip tendencies calculated for each fault. For $T_s$ > 0.3, the fault is deemed unstable, for $T_s$ <= 0.3 it is stable.





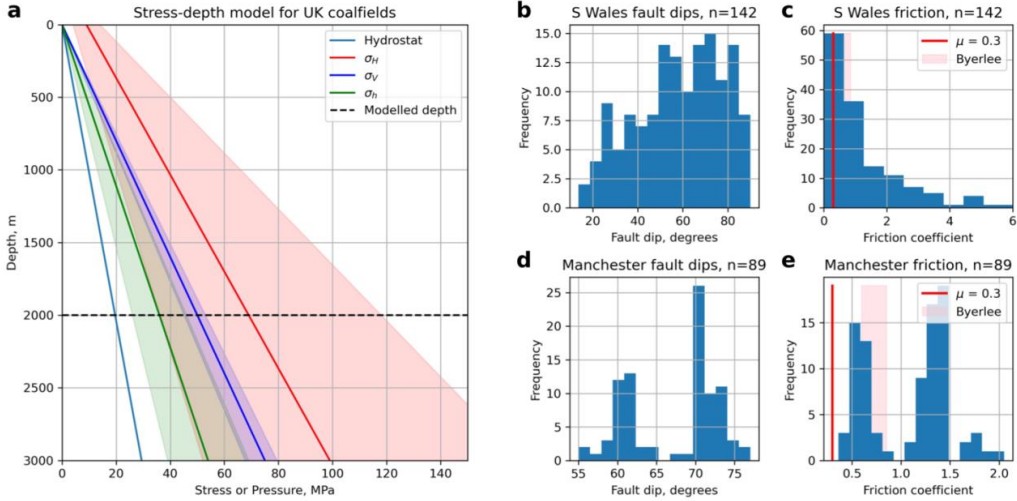

577

**Figure 15.** Constraints on input variables for the coalfield modelling of slip tendency. **a**. Stress-depth plot based on data from onshore UK (after Fellgett et al., 2018). Also shown is the modelled depth of 2 km. **b-e**. Histograms of fault dips measured cross-sections on published BGS 1:50k maps of South Wales and Greater Manchester, and calculated values of friction coefficients derived from these dips assuming Mohr-Coulomb failure. Byerlee friction ($\mu$=0.6-0.85) shown as shaded pink box. Modelled critical values of friction ($\mu$=0.3) shown by red lines.


| Variable | Mean | Standard deviation ($\kappa$ for Von Mises) | Units | Distribution | Comments |
|---|---|---|---|---|---|
| *South Wales coalfield $T_s$ model, depth=2 km* | | | | | |
| $\sigma_V$, vertical stress | 50.0 | 3.75 (5% of mean) | MPa | Normal | Lithostatic for depth of 2 km, assuming average rock density of 2500 kg m$^{-3}$ |
| $\sigma_H$, max. horizontal stress | 70.0 | 14.0 (20% of mean) | MPa | Normal | After Fellgett et al., 2018 |
| $\sigma_h$, min. horizontal stress | 35.0 | 3.5 (10% of mean) | MPa | Normal | After Fellgett et al., 2018 |
| Azimuth of $\sigma_{Hmax}$ | 160 | $\kappa$=200 | ° | Von Mises (circular Normal) | After Fellgett et al., 2018; Baptie, 2010; WSM, 2016 |
| Fault strike | - | - | ° | As mapped | Digitised from BGS Hydrogeology sheet |
| Fault dip | n/a | $\kappa$=25 | ° | Von Mises (circular Normal), truncated at 0 and 90 | Fitted to data taken from cross-sections on BGS 1:50k sheets 229-231, 247-249, 263, 263 |
| *Greater Manchester coalfield $T_s$ model, depth=2 km* | | | | | |
| $\sigma_V$, vertical stress | 50.0 | 7.5 (5% of mean) | MPa | Normal | Lithostatic for depth of 2 km, assuming average rock density of 2500 kg m$^{-3}$ |



| $\sigma_H$, max. horizontal stress | 70.0 | 14.0 (20% of mean) | MPa | Normal | After Fellgett et al., 2018 |
|---|---|---|---|---|---|
| $\sigma_h$, min. horizontal stress | 35.0 | 3.5 (10% of mean) | MPa | Normal | After Fellgett et al., 2018 |
| Azimuth of $\sigma_{Hmax}$ | 145 | $\kappa$=200 | ° | Von Mises (circular Normal) | After Fellgett et al., 2018; Baptie, 2010; WSM, 2016 |
| Fault strike | - | - | ° | As mapped | Digitised from BGS 1:50k sheets 84-86 |
| Fault dip | 60.0 | $\kappa$=200 | ° | Von Mises (circular Normal), truncated at 0 and 90 | Fitted to data taken from cross sections on BGS 1:50k sheets 84-86 |


**Table 4.** Distributions of input variables used to model slip tendency in the coalfields of South Wales and
Greater Manchester.
Predictions of conditional probability for fault slip have been calculated for all faults in both coalfields using
slip tendency as the chosen measure: in the absence of detailed pore fluid pressure constraints or estimates
of cohesive strength, it is hard to justify modelling the fracture susceptibility. Slip tendency provides a first
order estimate of fault stability. A quadratic response surface was constructed for each coalfield using the
full range of measured fault strikes and dips, and the input variable distributions listed in Table 4 and
constrained by the data in Figure 15. Monte Carlo simulations ($N_{MC}$=5,000) were run for each mapped fault
segment with the other input variables drawn from their respective distributions. Note that the principal
stresses used were the same for both coalfields, for a depth of 2 km (see Table 4), but the azimuth of sHmax
was varied to reflect the regional differences reported by other authors (Becker & Davenport, 2001; Baptie,
2010), and the recorded focal mechanisms.
Output CDFs for all faults in both coalfields are shown in Figure 16. For South Wales (N=3,408 faults),
approximately 46% of faults are predicted to have a 1 in 3 chance of being unstable (i.e., $T_s$ > 0.3, shown in
red), and 42% of faults are predicted to have a 1 in 10 chance of being unstable (shown in amber). For Greater
Manchester (N=3,453 faults), approximately 46% of faults are predicted to have a 1 in 3 chance of being
unstable (i.e., $T_s$ > 0.3, shown in red), and 54% of faults are predicted to have a 1 in 10 chance of being
unstable (shown in amber).

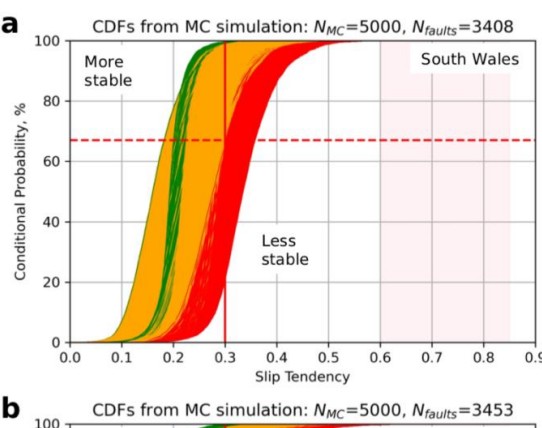

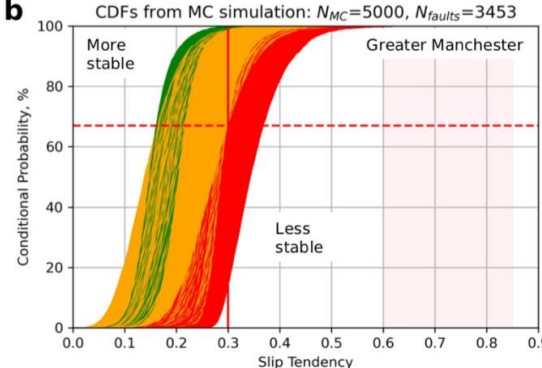


**Figure 16.** Output from the Monte Carlo modelling of slip tendency in UK coalfields. For slip tendency, more
stable faults skew towards the left (low $T_s$), less stable faults skew to the right (high $T_s$). **a**. CDFs of predicted
slip tendency for each mapped fault in South Wales. **b**. CDFs of predicted slip tendency for each mapped fault
in Greater Manchester. Colour coding of CDFs – red: >33% chance of exceeding threshold friction ($\mu$=0.3,
vertical red line), amber: >1% and <33% chance, green: < 1% chance. Range of Byerlee friction shown by pink
shading.

The results from the RSM/MC modelling shown in the CDFs are replicated in map view in Figures 17 and 18.
Each fault segment is colour coded using the same heuristic applied in the CDF: red faults have a conditional
probability of at least 33% of their slip tendency exceeding the chosen threshold value of fault rock friction
($\mu$=0.3), amber (orange) faults have a 1-33% chance, and green faults have a less than 1% chance of being
unstable.

For South Wales, the general pattern of the predictions is consistent with the recorded focal mechanisms
(Figure 17a). The most likely fault segments to slip (coloured red) are those oriented either NNW/SSE or N/S,
corresponding with one of the nodal planes in each of the focal mechanisms. Faults trending ENE/WSW, such
as the Neath Disturbance, are predicted to have low probability of slip in the modelled stress regime (green).
Note that the Swansea Valley Disturbance trends ENE/WSW as a fault *zone*, but the constituent fault
segments are variously oriented including elements that trend NE/SW, and these are marked in red (high
probability of slip). Blenkinsop et al. (1986) noted that this fault zone may in fact have a shallow dip at depth,
which is not covered by the dip distribution used in our modelling, so further work is required here. The
location with the most recorded events lies to the SE of Merthyr Tydfil, and this corresponds to an area with
many mapped faults trending NW/SE marked with a high probability of slip, and consistent with two of the
focal mechanisms.
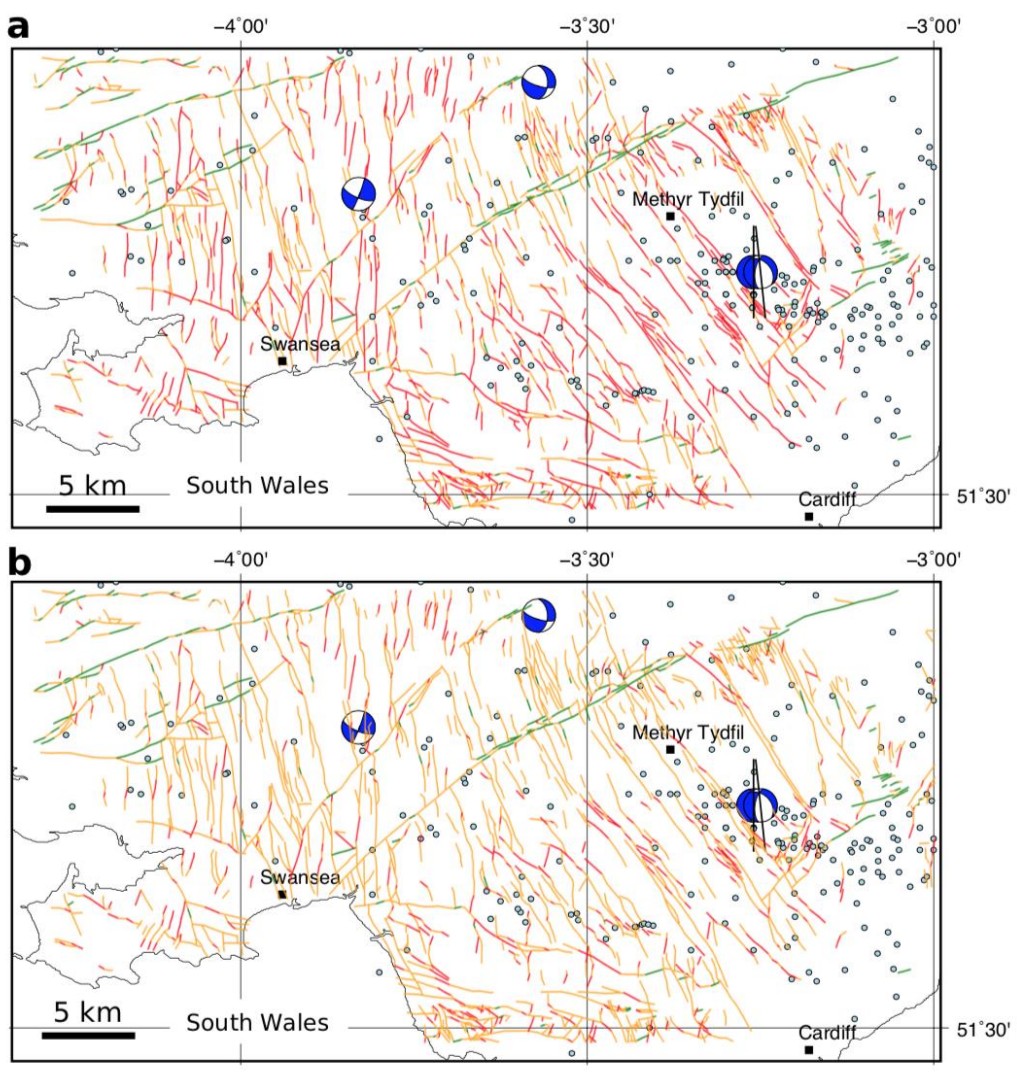

627

**Figure 17.** Output from the Monte Carlo modelling of slip tendency in South Wales coalfield. **a.** Colour-coded fault map showing conditional probability of slip for each mapped fault. This map shows the unweighted values, as shown on the CDFs in Figure 14a. **b**. Colour-coded fault map showing conditional *weighted* probability of slip for each mapped fault. The weighted probability is calculated by multiplying the probability from the CDF in Figure 14a by the normalised fault smoothness, ranging from 1.0 for a perfectly straight (i.e., smooth) fault, and tending to 0.0 for a rough fault. Colour coding of CDFs – red: >33% chance of exceeding threshold friction (μ=0.3), amber: >1% and <33% chance, green: < 1% chance.










For Greater Manchester (Figure 18a), the simulation suggests that many faults are likely to slip in the
modelled stress regime, even though the recorded seismicity is generally sparse. The exception is the area
of the 2002-2003 swarm near Manchester city centre. Here the recorded events coincide with mapped
surface faults trending WNW/ESE and predicted as likely to slip (red).

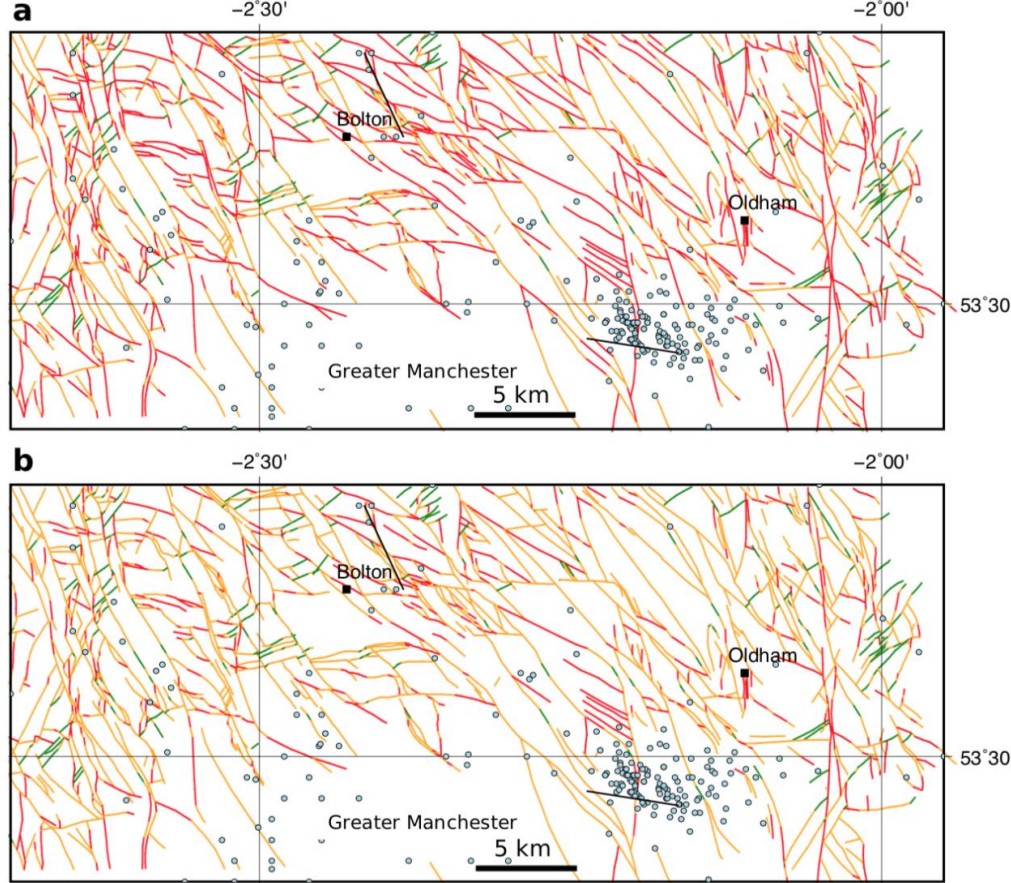


**Figure 18.** Output from the Monte Carlo modelling of slip tendency in Greater Manchester coalfield. **a**.
Colour-coded fault map showing conditional probability of slip for each mapped fault. This map shows the
unweighted values, as shown on the CDFs in Figure 14b. **b**. Colour-coded fault map showing conditional
*weighted* probability of slip for each mapped fault. The weighted probability is calculated by multiplying the
probability from the CDF in Figure 14b by the normalised fault smoothness, ranging from 1.0 for a perfectly
straight (i.e., smooth) fault, and tending to 0.0 for a rough fault. Colour coding of CDFs – red: >33% chance
of exceeding threshold friction ($\mu$=0.3), amber: >1% and <33% chance, green: < 1% chance.

**Discussion**
*Stress, pressure, and temperature*
The simulations described in this paper all critically depend on our knowledge of the *in situ* stress tensor. We
can constrain some of the components of this tensor better than others. The vertical stress ($\sigma_V$) is usually the
best constrained, a reflection of its derivation from the borehole density logs sampled at sub-metre
resolution. Our estimates of the horizontal stresses, $\sigma_{Hmax}$ and $\sigma_{hmin}$, remain poorly constrained. Even in cases



with relatively good data, e.g., from borehole leak-off tests (LOTs) and formation integrity tests (FITs), the
"data density" for these stress components is generally sparse (compared to $\sigma_V$), and we are stuck with
significant uncertainties. And these uncertainties matter, as shown by this study and previous work (e.g.,
Chiaramonte et al., 2008; Walsh & Zoback, 2016). The fundamental dependence of shear failure on
differential stress inherent in the Mohr-Coulomb failure criterion is reflected in the high ranking of stress
tensor components in the tornado plots shown in this study. Also, larger uncertainties in stress components
mean that the Andersonian regime may flip from the default "average" assumption to another orientation:
e.g., an apparently strike-slip regime may in fact include a significant proportion of normal fault possibilities
(>20% in the case of the Porthtowan Fault Zone shown here). One way to improve our knowledge of the
stress tensor, and especially the azimuth of $\sigma_{Hmax}$ would be to exploit richer catalogues of seismicity to
produce more focal mechanisms for natural or induced events. Most countries would benefit from better –
i.e., more widespread and higher resolution – continuous seismic monitoring. While this may be expensive
with top of the range broadband equipment, citizen science devices, such as the Raspberry Shake, offer a
low cost and viable alternative (Cochran, 2018; Anthony et al., 2019; Hicks et al., 2021; Holmgren & Werner,
2021). Our study shows how Raspberry Shake data are effective for computing focal mechanisms. Analysis
of more events would allow stress inversions to be performed on the data measured by these devices,
especially when they are combined in *ad hoc* arrays to improve signal to noise ratios.
Pore fluid pressures at depth are also poorly known, even for a country like the UK with a long tradition of
geological (and geophysical) science and rich history of mining and drilling into the crust. Most importantly,
our knowledge of measured *in situ* pore fluid pressures in and around fault zones is generally poor.
Theoretical predictions and model simulations abound, but direct measurements of this key parameter are
almost non-existent. We need to know the actual limits of pore fluid pressures in fault zones, and their likely
spatial and temporal variation over a fault plane throughout the seismic cycle. The situation is complicated
by the finer scale structure of fault zones. Fault zones in low porosity and/or crystalline rocks (such as granite)
can be divided into one or more narrow cores defined by fine grained fault rocks (gouges, cataclasites)
surrounded by wider damage zones of more or less fractured rock. Permeability may be low in and across
the core(s) and higher in the damage zones (Caine et al., 1996; Faulkner et al., 2010). In high porosity and/or
granular rocks (such as sandstone), fault zones may be simpler, with a fine grained fault rocks along narrow
fault planes forming an effective fluid seal (Wibberley et al., 2008) These differences in the physical
characteristics of the fault zones have consequences for the distribution of dynamic pore fluid pressures,
which remain poorly known in detail.
The work described in this paper has ignored the effects of temperature. However, thermoelastic stress may
be more important than poroelastic stress by a factor of 10 (Jacquey et al., 2015). In short, colder injected
water may increase the chance of slip on a given fault. In the UK, our knowledge of the subsurface
temperature field is increasing (Beamish & Busby, 2016; Farr et al., 2021), but we need more data, and again,
especially from faulted rocks.
*Faults*
An implicit assumption in all of the modelling performed in this paper (and many others) is that we know
something about the fault which may slip: i.e., we can only quantify risk on known faults. There will, in
general, be many more unmapped faults in the subsurface, and these may be the ones most likely to slip due
to a change in loading (of either *in situ* stress or fluid pressure). This is apparent in the maps for the coalfields
shown in this paper in terms of the relative lack of correspondence between the surface mapped fault traces
and the locations of recorded earthquakes. Some of this "mismatch" could be explained by the dip of the
faults measured at the surface, but not all. Moreover, there are areas of apparently intense surface faulting
and no recorded seismicity, and vice versa (recorded seismicity but no mapped surface faults). Some advance
could be made to address this problem with the recognition that each recorded seismic event documents a
fault plane, assuming that a double couple focal mechanism implies fault slip rather than dilation from dyke
emplacement or other mechanisms. And therefore the 3D position of each focal mechanism points to at least
part of a subsurface fault. The challenge then lies in mapping these seismic event fault planes into a viable
fault network. Better data (i.e., higher spatial resolution and extending to smaller event magnitudes) from





more dense arrays of seismometers would help with this task, as for the refinement of stress estimates noted
above.
*Rock properties*
The importance of good data on rock properties has been emphasised above, in the Worked Example for
fracture susceptibility and in the case study for the Porthtowan Fault Zone. In general, we need more and
better data on coefficients of friction and cohesive strength, especially for the target formations of
decarbonisation operations. Moreover, we need data for the intact *and* faulted rocks. We also need better
constrained correlations among rock properties. A widely used method in oil and gas is to derive estimates
of friction coefficient and UCS from wireline log datasets measuring porosity, slowness (velocity) or elasticity
e.g., Chang et al., 2006. However, as noted by these authors, the correlations are strictly valid only for the
specific formations tested in the laboratory, and even then, the uncertainties remain large. A further issue is
the tendency to average wireline log derived estimates over a depth interval, when for most sections of crust
this is the direction in which rock properties are expected to vary most rapidly. The Porthtowan Fault Zone
example above highlighted another issue: the relative impact of cohesion and friction on the predicted
stability depends on the magnitude of the cohesion in relation to the shear stress on the fault. For low
cohesion values, the constraints on friction become much more important. We need systematic
investigations of frictional behaviour at low cohesive strength. We need detailed systematic correlations
among rock properties, especially for faulted crystalline basement rocks.
Collecting more laboratory data is no panacea, evidenced by the well-aired concerns over how we up-scale
rock properties and behaviours from mm- and cm-sized samples to whole fault zones. But calibrations and
correlations from careful, systematic laboratory data remain the cornerstone of estimating the key *in situ*
values. An interesting new focus would be to explore the nature of the skewness in mechanical property
datasets: why should friction coefficients skew low, and cohesive strength skew high?
The utility of the Mohr-Coulomb criterion used in this paper is largely down to its mathematical simplicity,
i.e., linearity and only two parameters (friction and cohesion). Other criteria are perfectly viable and could
easily be added to the pfs Python code, but some other failure criteria lack a clear mapping between their
parameters and the mechanics of sliding on rock surfaces.
*Applicability of $T_s$, $T_d$ and $S_f$ for quantifying risk*
A valid question is to ask whether any of these widely used measures of fault stability are, in fact, useful in
practical terms at the scale of faults on maps. All three measures focus on the simplified mechanics of slip on
a specific fault plane, with a fixed orientation and with specific rock properties. But seismic hazard is not
isolated at the level of single fault planes. Faults occur in patterns or networks, more or less linked together.
Geometrical factors may be more important than the specifics of either the *in situ* stress or the rock
properties, at the scale of observation. The observational record shows that bigger fault zones are the sites
of bigger earthquakes, and they are also the locus of most displacement in a given network. Conversely,
smaller faults host smaller seismic events, and accrue less overall displacement (Walsh et al., 2001). To begin
to address this issue, we can weight the conditional probabilities of slip for a specific fault segment by a
dimensionless normalised factor derived from the total length of the fault: e.g., $w_{size} = l_s / l_t$ where $l_s$ is fault
segment length and $l_t$ is fault trace length. An alternative, but related idea, is that of the relationship between
fault smoothness (or inversely, roughness) and fault maturity, and therefore seismic hazard (Wesnousky et
al., 1988). The most seismically active faults are not only, or necessarily, the largest ones in their network,
but tend to be the smoothest or most connected, reflecting the coalescence of fault segments through time
and the removal of asperities through repeated slip events (Stirling et al., 1996). Therefore, we can weight
the conditional probabilities of slip by a dimensionless factor of smoothness: $w_{smooth} = l_{straight} / \text{sum}(l_s)$, where
$l_{straight}$ is the straight line length between fault end points, which is 1.0 for a perfectly smooth fault with all
segments parallel and connected, and tends to 0.0 for rough, complex fault traces. Examples of the effect of
these smoothness weightings applied to the conditional probabilities are shown in Figures 17b and 18b for
the UK coalfield faults. The net effect is to reduce the number of most risky faults (shown in red) by about
half. These approaches are the subject of further work and testing.



**Summary**
In this paper, we have described and explained the Response Surface Methodology and shown how it can be
combined with a Monte Carlo approach to generate probabilistic estimates of fault stability using published
measures of slip tendency, dilation tendency and fracture susceptibility. Simulations show that a quadratic
response surface always generates a better fit to the input variables in comparison to a linear surface, at the
cost of larger matrices (more computer memory) and longer run times. Worked examples to calculate $T_s$ and
$S_f$ with synthetic input distributions show how the quadratic response surfaces vary for each input parameter.
For slip and dilation tendency, the primary dependence is (as expected) on the maximum differential stress,
and therefore the maximum and minimum principal stresses of the *in situ* stress tensor, with a lesser
dependence on the fault orientation. For fracture susceptibility, the situation is more complex: if cohesion is
relatively high, $S_f$ is mainly dependent on the *in situ* stresses and cohesion. But if cohesion is low – quite likely
in fault zones – then the dependence of $S_f$ on friction is much more significant. This is a key finding: the
relative sensitivity of the input variables on the response surface varies with the absolute value of the
variables.
Sensitivity tests were used to assess how the shapes of different input distributions affect the predictions of
fault stability. Varying the spread of symmetric (normal, Gaussian) distributions of input variables has a
significant effect on the predictions, and this mirrors the reality of uncertainties in, for example, the principal
stresses in a standard geomechanical analysis. As noted above, the vertical stress is often well constrained
and has a lower relative standard deviation (say, 5% of the mean) than either the maximum or minimum
horizontal stresses (typically 15-20% of their mean value). The shape and spread of skewed (asymmetric)
distributions of rock properties (friction and cohesion) is also important. The direction of skewness is
described by the sign of the parameter $\alpha$ for the skewed normal distributions used in this paper to model
variations in rock properties. Friction is modelled with a negative skewness towards lower values, whereas
cohesion is modelled with positive skewness towards higher values, but systematic laboratory data are
needed to verify these assumptions. This will require a statistically significant number of repeat tests for each
property on quasi-identical samples of the same rock.
Case studies of three different locations demonstrated how a probabilistic approach can provide a useful
assessment of fault stability, including which of the input variables are the most important for a given
combination of *in situ* stress, fault plane orientation and rock properties. This then enables greater focus on
improving the estimates of the key variables, and the relationships between them. For the Porthtowan Fault
Zone in Cornwall, the modelling in this paper shows that we need more data for, and a better understanding
of the relationship between, coefficients of friction and cohesive strength, especially at low values of friction
(i.e., less than the Byerlee range of 0.6-0.85) to be expected in fault zones. For the coalfields in South Wales
and Greater Manchester, model outputs show how predictions of fault stability can be weighted by a simple
index of fault smoothness to begin to allow for the effects of geometrical weakening within the fault system
as whole, rather than focusing on each individual fault plane taken in isolation.
It's obvious that uncertainty in the input parameters must translate into uncertainty in the output
predictions. By combining a Response Surface Methodology with a Monte Carlo approach to the
quantification of fault stability, we can explore, understand, and quantify how differing degrees of
uncertainty among the input parameters feed through to uncertainty in the predicted stability measure.
Response surfaces and tornado plots can help to identify which parameters are the most important in a
particular analysis. Given our current state of knowledge of stress, fault orientations and fault rock
properties, probabilistic estimates and iterative modelling are useful approaches to begin to de-risk the
energy transition. Free, open source software to perform these analyses, such as the Python package pfs,
can help to encourage their wider adoption and further refinement ("given enough eyeballs, all bugs are
shallow"; Raymond, 2001). The deployment of abundant and relatively low-cost citizen science seismometers
(e.g., Raspberry Shakes) could synergise two critical issues: the wider involvement of the public into open
science debates about risk and the simultaneous collection of better data to constrain the local stress field.
The energy transition and decarbonisation are urgent and essential tasks: we will only be successful if we
manage to balance public perceptions of risk with the technical challenges inherent to the exploitation of
faulted rock.




**Appendix A – Dilation tendency plots**

For completeness, we include the analysis of dilation tendency ($T_d$) for the same synthetic input dataset used to calculate slip tendency ($T_s$) – i.e., input variable distributions taken from Table 2.

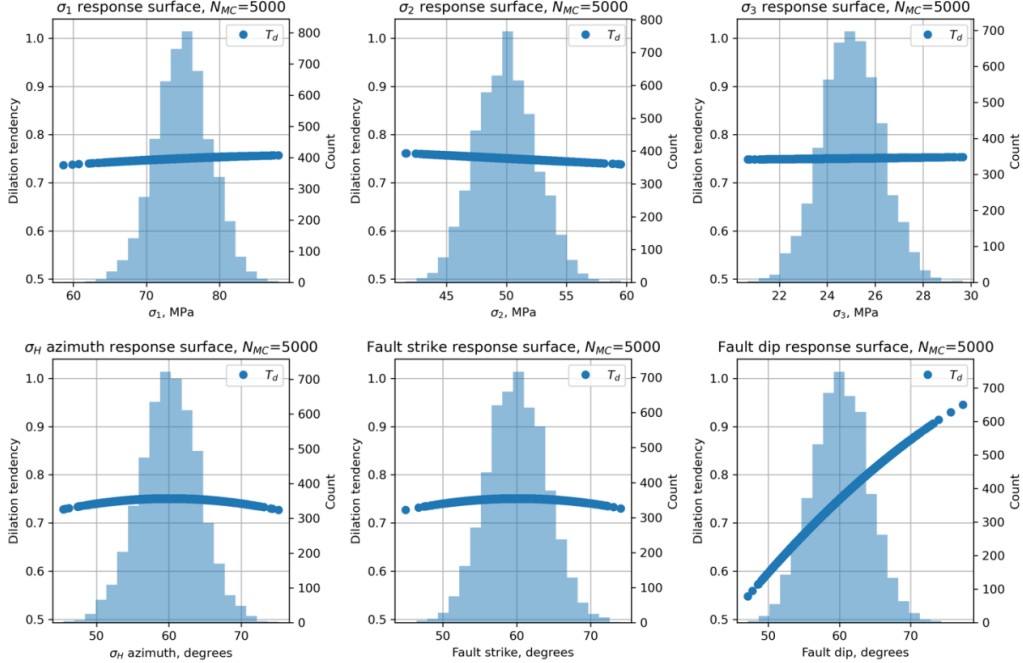

**Figure A1.** Histograms of input variables used to calculate dilation tendency $T_d$ for the synthetic distributions shown in Table 2.

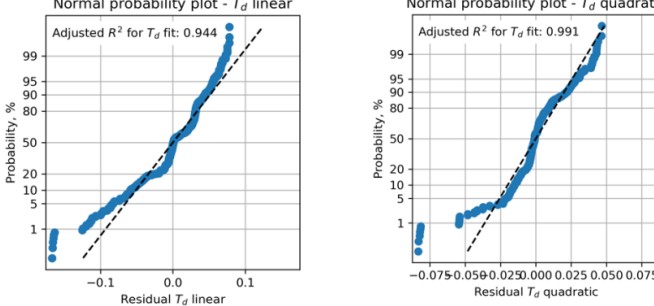

**Figure A2.** Residual plots for linear and quadratic response surfaces for dilation tendency using synthetic data. The quadratic fit has a higher value of the adjusted $R^2$ parameter and is therefore deemed better in this case.





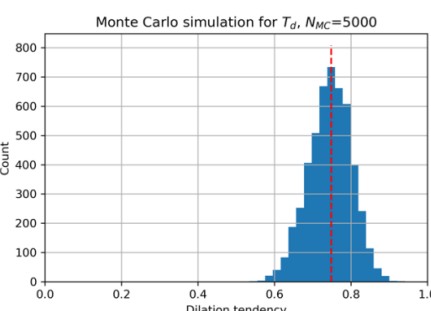 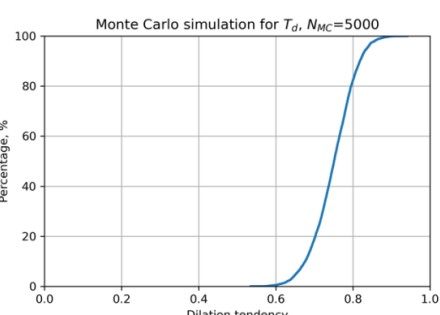


**Figure A3.** Output from Monte Carlo simulation ($N_{MC}$=5,000) of dilation tendency calculated using a quadratic
response surface from synthetic input data. **a**. Histogram of calculated dilation tendency values, in this case
showing a quasi-normal distribution with a mode of ~0.75. **b**. Cumulative distribution function (CDF) of
calculated dilation tendency values, showing the range in values from ~0.5 to ~0.9.

**Code availability**
https://github.com/DaveHealy-github/pfs

**Data availability**

**Author contribution**
DH – 80%, SH – 20%. DH originated the study, wrote the code, ran the models. SH contributed seismology
data and expertise, and contributed to the writing of the text.

**Competing interests**
The authors declare that they have no conflicts of interest.

**Acknowledgements**
DH first presented the core ideas in this paper at the Tectonic Studies Group AGM in Cardiff in 2014, and
enjoyed discussions there with Dr Jonathan Turner (RWM Ltd). Thanks to former PhD student Dr Sarah
Weihmann (now at BGR) and co-supervisor Dr Frauke Schaeffer (Wintershall DEA) for discussions about using
oil industry wireline log data for quantifying geomechanical models. GMT (Wessel et al., 2013) was used for
the maps. SciPy (Virtanen et al., 2021), Numpy (Harris et al., 2020), and matplotlib (Hunter, 2007) were used
for the Python pfs code and Allmendinger et al. (2012) for various geomechanical and geometrical algorithms.

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
