# Peer review of "De-risking the energy transition by quantifying the uncertainties in fault stability 1"

_Solid Earth, 2021_

## Author Response (ED1)

**Response to reviewers comments**

We thank the reviewers for their comments. Below we detail our responses (in red) to each point raised.
Changes to the revised ms are listed in blue.

**1. Community Comment by Bill Lanyon**

Fault plane properties for United Downs case

The cohesion and frictional properties used describe the strength of intact rock specimens (from Elliott
1984). It seems very unlikely that a fault / fracture plane would have significant cohesion. Wouldn't it be
more reasonable to assume only frictional strength for fault planes?

In general we agree, a fair point. However, in terms of the overall model for fracture susceptibility, we think
it is useful to include cohesion as a variable. We can run models for cohesion set to 0 to address this issue.

In terms of the fault zone itself, we haven't seen cuttings or core from the boreholes, nor outcrops of the
PFZ. So we don't know if this fault zone is best characterised in terms of fault core + fault damage, with
gouge and cataclasite in the core; or, if it is better considered as a fracture corridor of pre-existing joints
and veins. In the latter case, we might expect some, albeit small, cohesive strength (perhaps a few kPa). In
the former case, "strength" would be better captured as frictional strength. But there would then be a
dynamic aspect to that, beyond the simple Mohr-Coulomb analysis we have used (e.g., experimental
evidence for frictional strength increasing with longer hold times between slip events).

In the case of zero cohesion, the comments we make still stand: that our knowledge of friction coefficients,
and especially their statistical distribution - skewed high or low - could be better.

I agree that it's potentially useful to keep cohesion in the formulation but using the intact rock values is just
going to significantly overestimate the fault plane strength at low normal stress. I don't think much data
has been released from the UD deep borehole yet, but for mechanical properties there's only side-wall core
and chippings both of which might be quite limited in providing frictional properties for fault planes at
seismic scales. The image logs probably give some guide as to the structure. So while it's nice to get more
data there will still be a lot of expert judgment on fault plane frictional properties that go into a pfs model
and "caution" is probably going to lead to using zero cohesion in many cases.

OK, agreed.

There is now some published information on the PTF at the UD site in a paper from Reinecker et al. in
Geothermics.

We have read this paper and cite this in our ms.

Lines 394 and 397.

**Response to Reviewer 1 – Jonathan Turner**

We thank the reviewer for their comments (repeated below in black) and provide detailed responses below
(in red).

This paper addresses a topic of general societal interest; it is well written, carefully explained, thoughtful.
The paper highlights the importance of several fault zone processes which are previously known about but
this study provides fresh perspective e.g. the role of uncoupled fluid pressure, coupled fluid pressure
(poroelasticity), the frictional properties of fault rocks (gouge, cataclasites), the importance of optimally
constraining in situ stress measurements, etc. In fact I think poroelasticity deserves wider discussion here
and elsewhere because it may be the key to understanding the unpredictability of induced seismicity.

We are pleased that the ms is considered to be well written, carefully explained and thoughtful.

I have few substantive comments and recommend that this paper be published.

Thank you.

1. It may be a bit too long with a little too much space devoted to explaining the method (or perhaps
consider cutting one of the two synthetic case studies).

OK, fair point (also made by Reviewer 2 – Anon). We will delete the Manchester coalfield case study and
focus on United Downs deep geothermal for fracture susceptibility and South Wales coalfield shallow
geothermal for slip tendency.

Section 3.2

2. Rangely oil field, Colorado is another good example to cite of a case study which showed a critical
threshold in fluid pressure, above which seismicity was induced and below which it was absent (sorry, I
don't have the reference but I think it was in the 1970s).

Good point. We will include this seminal study in our background and/or discussion.

Line 62.

3. The Townend & Zoback dataset is intriguing but in my experience very difficult to apply to development
projects. What I mean is that is it hard to demonstrate that critically stressed faults are
conductive/transmissive/higher perm, at least in as clearly as the T & Z dataset shows they should be.

Agreed. But tackling this is beyond the scope of our ms.

4. I got confused by the difference between slip tendency and friction coefficient – in words, friction
coefficient is the ratio of shear force to tractional force at the moment of failure. So I then thought it's no
surprise that your modal slip tendency in the first case study is 0.56 because that's the inverse tan of
~30degree which is an 'average' angle of internal friction for compacted rocks. I would find it useful if this
point could be explained in slightly more detail.

Our understanding is that slip tendency is a feature of the stress field and the fault plane orientation (shear
stress/normal stress), whereas friction is a rock property (an empirical measurement from laboratory
tests). So if slip tendency exceeds friction, then a fault slips; if slip tendency is less than friction, there is no
slip.

The friction coefficient will vary for different lithologies, different fault rocks, slip rates, etc. So for recently
formed faults in the present day stress field, slip tendency ought to be about 0.6-0.85 (Byerlee). But that
does not have to be the case for "old" faults formed under different stress states, not least because the
orientation of the present day in situ stress is not the one at the time of faulting (e.g., Carboniferous or
Permian in the case of the UK coalfields).

**Response to Reviewer 2 – Anonymous**

We thank the reviewer for their comments (repeated below in black) and provide detailed responses below
(in red).

First of all, I would say that the authors are top scientists in this field and accordingly, the idea and the
methodology reported in this paper seem to be very promising. Moreover, for people like me with a
prevalent geological background, the pure statistical part of the paper can be hard to be read just because
of the background.

Thanks. One key aim of our ms (see lines 52 – 59) is to explain the underlying theory and statistical
background to the Response Surface Methodology for just these reasons. And according to Reviewer 1, it is
"well written, carefully explained and thoughtful".

No change.

However, the geological data seems to be, in my opinion, poorly exposed here and the statistics are
sometimes completely detached from the geological data making this paper quite difficult to be read from
a Solid Earth reader.

We don't understand what is meant by "geological data seems to be … poorly exposed here". We have
used the publicly available geological data for each case study, and cited all the sources.

Also, we do not understand the comment "the statistics are sometimes completely detached from the
geological data". In the absence of complete certainty in the available data, we have used specific statistical
distributions to model the consequences of uncertainty.

No change.

Generally speaking, the paper faces a very interesting problem, and the method is innovative and very
exciting. As far as I can see the methodology is new and for this it must be tested and verified yet. The
authors attempt to do this by presenting two case studies with the aim to show "how combined RSM/MC
approach can be used to estimate the probability of slip on one or more faults".

We agree that this is interesting, innovative and exciting.

No change.

However, the two cases are not very well constrained in terms is of boundary conditions making the
probability estimation quite confused.

We do not understand what the reviewer means by "boundary conditions". We are not conducting a
numerical modelling analysis of a fixed spatial or temporal domain, e.g., of tensor fields or conservation
equations using finite differences, and therefore the notion of formal boundary conditions is misplaced, in
our opinion.

Our analysis, described in the first two sections, focuses on modelling the consequences of uncertainties in
all of the possible input parameters involved in the quantification of fault stability (using either fracture
susceptibility and slip tendency). As such it is a direct extension and development of the work presented by,
for example, Chiaramonte et al. (2008) and Walsh & Zoback (2016). We do not think the probability
estimation is "quite confused" (cf., comments by Reviewer 1).

No change.

Moreover, the two performed analyses (Porthtowan Fault Zone in Cornwall, UK and Coalfields in South
Wales and Greater Manchester, UK) differ in so many aspects and, more importantly the presented results
are different in terms of delivered outputs. This make the reading quite confusing and at the end of the
paper I got lost about the point that the authors would like to address. In my opinion to test a new
methodology we should apply this in areas where data are known as much as possible to see if the model
prediction are reliable. In this case since the two areas are poorly constrained, this exercise is difficult to be
followed and the results even more difficult to be understood.

We agree the case study areas are different, and the chosen modelled outputs are also different. This is all
deliberate. Our intention is to demonstrate the scope of the method (combined RSM and MC) to make
useful predictions about fault stability in terms of fracture susceptibility (United Downs) and slip tendency
(coalfields) in the face of uncertainty.

As noted above in Response to Reviewer 1, we will remove the Manchester coalfield case study to reduce
the length of the ms. We hope this makes it easier to appreciate the differences – and more importantly,
the value in those differences – in the two case studies.

In relation to "we should apply this in areas where data are known as much as possible to see if the model
prediction are reliable": we know of no such datasets. In the case of United Downs – arguably one of the
best constrained sites involved in geothermal energy – all of the data remain uncertain (to varying
degrees), and this is one of our key points: even for areas with apparently "good" data, we argue that the
existing uncertainties are significant and have consequences.

Section 3.2 has been reduced.

The discussion paragraphs more than discuss the results present a list of what we should know to better
assess the seismic risk and the main message seems to be that we would need to know a lot of things. I can
kind of agree with this but, once again, this makes the main message of the paper more confused.

We disagree with this comment and agree with Reviewer 1 that the ms is "well written, carefully explained
and thoughtful".

I strongly suggest the authors to simplify the paper in two ways.

1.   Try to organize a sort of sensitivity analysis of the involved parameters in a more
  structured and ordered way in order to facilitate the reader
2.   Focus in one area and compare the results with something actually observed.

For the first point, sensitivity analyses are already included in the worked examples and in the case studies;
for example, we use CDF plots to explore the absolute sensitivity to selected parameters and we use
tornado plots to rank the relative sensitivities (see Figures 4, 5, 7 & 8).

For the second point, we think the reviewer might have missed the point. We know of no site or area
where the observations are known perfectly, i.e. with 100% certainty.

No change.

I think that we all agree that there are many topics related to the risk assessment (fault length, roughness,
friction, fluids, background seismicity, regional strain rate, and many many others) but in doing this exercise
authors must clearly state the assumption and critically analyze the results. In this paper I had the
impression that speaking about the many variables we lose the point of the paper, I would say that
sometimes less is more.

We have stated the assumptions used throughout (e.g., Mohr-Coulomb failure), and we critically analyse
the results through detailed statistical analysis of the outputs. One of our main aims, clearly listed in the
Introduction, is to provide a clear and detailed explanation of the method (in our opinion, so far lacking in
previous publications using similar methods). This entails some detailed and "careful explanation"
(Reviewer 1).

No change.

Minor points:

I am not so convinced about the statistical discussion that is sometimes too focused on the pure statistics
and few on the geology behind. For example, can we find a geological meaning to the "asymmetrical or
skewed" distribution of some parameters?

This is one of the issues raised by our ms, and clearly discussed! By trying to accommodate the fact of
uncertainty in all input parameters – stresses, orientations and rock properties – we are faced with making
choices about the nature (shape) of their distributions. We clearly state that there is currently insufficient
published data for many of these parameters – especially some critical ones such as cohesion and friction –
to find any "geological meaning".

No change.

I Am not expert on Response Surface Methodology (RSM). However, the paragraph Statistical analysis of
geomechanical fault stability start with a discussion on the governing equations for RSM following a quite
long description that ends with the definition of Ts by meaning of the very well-known direction cosines
(e.g. Ramsay and Lisle 2000). In other word I can't really see why the authors need introducing the RSM
theory to infer the Ts definition.

The reviewer has perhaps missed the point. We are not "inferring" the Ts definition. The equations for Ts
are given in their full format (i.e., in terms of direction cosines) to highlight one of the key issues: there are
8 input parameters, and they are all, in general, uncertain. This is picked up in the succeeding paragraph
(line 221 in the original ms). We need to show the full equation for Ts before we make this crucial point.

No change.

A lot of acronymous BGS, CDF, are used but not defined. Even if they are quite easily understandable, this
gives the impression of a lazy writing

We presume the reviewer means "acronyms". BGS is the British Geological Survey – we will add a definition
for that. CDF is defined on first use, on line 138.

BGS is now defined on first use in the main text, line 408.

The discussion on the relationship between fault length and events magnitude starts with this and ends
with discussing the relationship between fault length and number of events. I would say that the two
(maximum magnitude and number of events) are surely correlated but they are not the same thing.

Agreed. But we do not say they are the same thing.

No change.

Line to line comments:

Line 228 I would say that fluid pressure also influences Ts (e.g. De Paola et al., 2007)

We strongly disagree. Pore fluid pressure plays no part in the formal definition of slip tendency (Ts) – see
Morris et al., 1996. Moreover, the influence of pore fluid pressure on the potential for failure is better
understood in terms of fracture susceptibility – i.e., the pore fluid pressure increase needed to drive the
stress state on the fault to failure (Streit & Hillis, 2000).

No change.

Line 239 is CDF the cumulative distribution function? Authors should state this somewhere.

Yes it is. It is defined on first usage, on line 138 of the original ms.

No change.

Line 326 alfa has been not defined

Definition for alpha ($\alpha$) will be added.

Now defined on first use, line 327.

Line 698 Why these may be the ones most likely to slip?

We are highlighting the *possibility* that unmapped (i.e., unknown) faults *may* be most likely – due to all the
factors discussed in this paper. The point is about unmapped faults, or so-called "known unknowns".

No change.

Line 700 Some of this "mismatch" could be explained by the dip of the faults measured at the surface, but
not all. What the author mean here?

We mean that the surface traces of the faults shown on our maps may not coincide with their extension at
depth, e.g., for faults that dip at less than 90 degrees. This could explain some of the apparent mismatch
between the recorded earthquake locations plotted on the map relative to the surface traces.

No change.

Line 742 The observational record shows that bigger fault zones. I would say that there are a lot of physical
reasons behind this. Moreover, empirical relationships such as those suggested by Wells and Coppersmith
1994, or Leonard 2010 should be cited here.

Thanks for these suggestions. We will add these papers.

Line 810.

Subsequent comments and replies...

I read the answers to my comments, and I have to say that I really hope that the Editor and all the SE
readers will find the whole paper "well written, carefully explained and thoughtful" . I still think that some
parts should be improved, however I just reported my suggestions hoping to help.

In any case, I would like just to comment on the answer regarding Ts dependence on fluids.

The answer was:

We strongly disagree. Pore fluid pressure plays no part in the formal definition of slip tendency (Ts) – see
Morris et al., 1996[…].

In the Morris et al paper Ts is defined by Tau/Sigma. Sigma are, generally speaking, the principal stresses
that might be interpreted as fluid pressure independent because effective stresses are not mentioned.
However, in the same paper, Morris et al., 1996 calculated the Ts for the Yucca Mountain area and, while
setting the input sigma, the literally write:

[…] to a depth of 5 km and assuming an average rock density of 2.7 g/cm3, s1 = 133 MPa, s2 = 5 88–108
MPa, and s3 = 5 63–72 MPa. Assuming a water-table depth of 600 m (Stock et al., 1985), and
interconnecting permeability hydrostatic pressure at 5 km will be 43 MPa. Thus, effective principal stresses
would be: s1=vertical=90 MPa, s2=N258E–N308E = 45–65 MPa (50%–72% of s1), and s3 = N608W–N658W
= 20–29 MPa (22%–32% of s1), at 5 km beneath Yucca Mountain.

Please note that the effective stresses are those used by Morris et al., in their calculation of Ts (Figure 3).
This is also confirmed by Lisle and Srivastava, 2004 that literally write: "If pore-fluid pressures are involved,
then the stresses should be considered effective stresses."

If effective stresses should be used, Ts would change with changing Pf, also because Tau is Pore-pressure
independent. I would say, thus, that I "strongly" believe that Ts does depend on Pf.

What can be independent from Pf is the Ts/Tsmax ratio (defined as "T's" by Lisle and Srivastava, 2004).
However, Ts and not T's is investigated in the present paper by Healy and Hicks.

Thanks again for the comments.

We agree that the formal definition of slip tendency does not include pore fluid pressure. The question
then is: is it useful to modify the normal stress term by subtracting the pore fluid pressure to get an
'effective normal stress', and an 'effective slip tendency'.

In our opinion, the power of the original definition of Ts is how it can be related to the friction coefficient at
the fault surface. That is, the slip tendency, a function of the stresses on the fault plane, can be compared
to the rock properties (the friction coefficient), and an assessment of stability can be made. It is not clear
how this works for 'effective' terms. Effective friction?

Therefore, to clearly separate potential frictional processes from hydraulic (pore fluid pressure) processes,
we believe it is better to keep the original definition of slip tendency, and use fracture susceptibility as an
index of stability under effective pressure/stress.

No change.

[revised manuscript text omitted]